# Expected Return Causes Outcome-Level Mode Collapse in Reinforcement Learning and How to Fix It with Inverse Probability Scaling

**Abhijeet Sinha** [1]   **Sundari Elango** [2]   **Dinabo Liu** [1]

## Abstract

Many reinforcement learning (RL) problems admit multiple terminal solutions of comparable quality, where the goal is not to identify a single optimum but to represent a diverse set of high-quality outcomes. Nevertheless, policies trained by standard expected-return maximization routinely collapse onto a small subset of outcomes, a phenomenon commonly attributed to insufficient exploration or weak regularization. We show that this explanation is incomplete: **outcome-level mode collapse is a structural consequence of the expected-return objective itself**. Under idealized learning dynamics, the log-probability ratio between any two outcomes evolves linearly in their reward difference, implying exponential ratio divergence and inevitable collapse—independent of the exploration strategy, entropy regularization, or optimization algorithm. We identify the source of this pathology as the probability multiplier inside the expectation and propose a minimal correction: **inverse probability scaling**, which removes outcome-frequency amplification from the learning signal, fundamentally changes the learning dynamics, and **provably yields reward-proportional terminal distributions**, preventing collapse in multimodal settings. We instantiate this principle in Group Relative Policy Optimization (GRPO) as a drop-in modification, IPS-GRPO, requiring no auxiliary models or architectural changes. Across different reasoning and molecular generation tasks, IPS-GRPO consistently reduces outcome-level mode collapse while matching or exceeding baseline performance, suggesting that correcting the objective rather than adding exploration heuristics is

key to reliable multimodal policy optimization.

## 1. Introduction

Reinforcement Learning (RL) is increasingly applied to areas where the objective is not to identify a single optimal behaviour, but rather represent a diverse set of equally high quality solutions. Such outcome-multimodal problems arise naturally in applications such as Large Language Model (LLM) post-training, (Ouyang et al., 2022) program synthesis (Simmons-Edler et al., 2018), molecular discovery (Park et al., 2025) and combinatorial generation tasks (Grinsztajn et al., 2023). In such domains, multiple terminal outcomes achieve comparable reward, and it becomes critical for the RL agent to maintain coverage across these alternatives instead of converging on a single solution (Romera-Paredes et al., 2024; Castro et al., 2025; Novikov et al., 2025). However, standard RL objective proves unable to capture the equally valid alternatives, often collapsing onto a narrow subset of outcomes (Hu et al., 2024; Gao et al., 2023). This phenomenon, referred to as *outcome-level mode collapse* (GX-Chen et al., 2025), yields high expected reward but offers poor coverage of the solution space leading to repetitive or homogenized output. The most prevalent explanation for this behavior is given as insufficient exploration (Song et al., 2025; Wang et al., 2025), suboptimal regularization (Wang et al., 2023) or even poor choice of hyperparameters. Thus, the strategies proposed to overcome this, are also on these lines. However, these approaches only delay collapse and do not address the root cause of the phenomenon. Thus, collapse reappears as training progresses and agents still fail to capture the global structure of the reward landscape.

In this work, we show that existing explanations are fundamentally incomplete and demonstrate that outcome-level mode collapse is a *structural consequence of expected-return maximization itself*. Even under ideal conditions with perfect exploration, and no optimization noise, maximizing expected return induces a rich-get-richer feedback loop at the level of terminal outcomes. Outcomes that are sampled slightly more frequently receive disproportionately larger updates, causing probability ratios between outcomes to drift and ultimately collapse onto a small subset of high-reward outcomes. Thus, we suggest that preventing mode

[1]National University of Singapore, Singapore [2]IIT Madras, India. Correspondence to: Abhijeet Sinha <abhijeet@nus.edu.sg>, Dianbo Liu <dianbo@nus.edu.sg>, Sundari Elango <sundarielango95@gmail.com>.

*Proceedings of the 43rd International Conference on Machine Learning*, Seoul, South Korea. PMLR 306, 2026. Copyright 2026 by the author(s).

collapse requires modifying the learning signal. Motivated by this, we introduce a simple and principled correction: *inverse probability scaling*. Instead of weighting each trajectory's contribution by its probability of occurrence, we rescale terminal rewards by the inverse probability of the corresponding outcome. This removes the probability multiplier inherent in expected-return optimization and hence, fundamentally alters the learning dynamics. Under this correction, the policy converges to a reward-proportional distribution over outcomes: $\pi_i \propto r_i$, thus the agent is better able to reflect the reward landscape. We instantiate this principle as *Inverse Probability Scaling*, a drop-in modification to group-based RL algorithms. We propose a modification to the GRPO algorithm (Shao et al., 2024), by applying this priniciple and term this algorithm as IPS-GRPO. IPS-GRPO requires no auxiliary models, and is fully compatible with existing GRPO pipelines, including PPO-style clipping and KL regularization. We show theoretically and demonstrate empirically that IPS-GRPO consistently mitigates mode collapse across a range of outcome-multimodal benchmarks, including controlled grid environments, structured hypothesis-space inference (HypoSpace), and molecular discovery with chemical language models.

The main contributions of this work are:

**Problem characterization:** We identify outcome-level mode collapse as a structural consequence of expected-return maximization in outcome-multimodal RL.

**Theoretical analysis:** We show this collapse arises from frequency-weighted learning dynamics rather than insufficient exploration or regularization.

**Algorithmic contribution:** We propose IPS-GRPO, a modification to GRPO that corrects outcome-frequency bias via inverse probability scaling, yielding stable outcome-level distributions.

**Practical applicability:** IPS-GRPO requires no auxiliary models, adds negligible overhead, and integrates directly with existing GRPO pipelines.

## 2. Related Work

### 2.1. Mode Collapse in RL and Loss of Diversity

Mode collapse was first observed in Generative Adversarial Networks (GANs) (Goodfellow et al., 2020) where the generator covers only a few modes of the data distribution resulting in samples that lack diversity (Kossale et al., 2022). An analogous collapse occurs in RL at the level of policies where even though multiple nearly or equally rewarding trajectories exist, the learned policies often collapse on to a narrow subset of them. This results in reduced policy entropy and loss of diversity in solutions, and has been documented across multiple domains. These failures have motivated a growing body of work that address this collapse by encouraging exploration through a variety of techniques like entropy regularization (Haarnoja et al., 2018; Cheng et al., 2025; Ahmed et al., 2019), or by increasing entropy in the training data (Zhou et al., 2025; Cui et al., 2025), imposing KL constraints between successive policies (GX-Chen et al., 2025; Rietz & Stork, 2023). Other methods modify rewards via shaping (An et al., 2025), intrinsic motivation (Davoodabadi et al., 2024), or exploration bonuses (Song et al., 2025) to bias learning toward underexplored regions of the state space.

### 2.2. Flow-based methods

A complementary line of work addresses diversity collapse by explicitly targeting distributions over terminal outcomes rather than maximizing expected return. Generative Flow Networks (GFlowNets) enforce flow-balance constraints to sample terminal states proportionally to reward, thereby avoiding mode collapse by construction (Bengio et al., 2023). Flow-based RL methods adapt these ideas to RL settings, using flow conservation or matching objectives to shape terminal distributions (Zhu et al., 2025). Energy-based models pursue a similar goal by defining unnormalized target distributions over outcomes and drawing diverse samples via MCMC or amortized inference (Chao et al., 2024). While effective at preserving diversity, flow-based and energy-based approaches typically rely on specialized objectives, auxiliary networks, learned normalization constants, or non-standard training pipelines. In contrast, our approach operates within standard KL-regularized policy-gradient frameworks and corrects outcome-level collapse by modifying reward attribution rather than enforcing explicit distributional constraints, preserving compatibility with existing RL pipelines.

### 2.3. Collapse in LLM post-training

Outcome-level collapse has become increasingly apparent in RL from human feedback (RLHF) and LLM post-training (Ouyang et al., 2022; Bai et al., 2022). Empirical studies report reduced response diversity, repetitive generations, and convergence toward narrow stylistic or semantic modes during reward optimization, even when multiple valid responses exist (Kirk et al., 2024; Slocum et al., 2025; Padmakumar & He, 2024). Similar effects have been observed in preference-based fine-tuning, where optimization favors high-probability responses preferred by the reward or preference model (Rafailov et al., 2024; Gao et al., 2023). In practice, collapse is commonly mitigated through strong KL regularization (Ouyang et al., 2022; Ziegler et al., 2020), reward mixing, early stopping (Stiennon et al., 2020), or decoding-time heuristics such as nucleus sampling and temperature scaling (Holtzman et al., 2020). Several works attribute these failures to reward misspecification, reward hacking, or excessive KL pressure (Amodei et al., 2016; Ouyang et al., 2022; Rafailov et al., 2024; Ziegler et al.,

2020). While these factors contribute, they do not fully explain why collapse persists even under careful tuning. Our work offers a complementary perspective: outcome-level collapse arises naturally from expected-return optimization over terminal outputs, and correcting this feedback at the level of reward attribution directly targets response homogenization.

# 3. Why Outcome-Level Mode Collapse is Inevitable under Expected Return

We show that outcome-level mode collapse is not an artifact of poor exploration, finite samples, or optimization noise, but a *structural property* of the expected-return objective itself. Even under idealized learning dynamics, maximizing expected return induces monotonic divergence of outcome probability ratios, forcing collapse onto a small subset of outcomes.

We consider episodic RL with reward only at termination. Each trajectory $\tau$ deterministically induces a terminal outcome $o = \phi(\tau) \in \mathcal{O}$, and reward depends only on the outcome: $r(\tau) = r(o)$. A policy $\pi_\theta$ induces a distribution over outcomes $p_\theta(o) = \Pr_{\tau \sim \pi_\theta}[\phi(\tau) = o]$. The standard objective is to maximize expected return,

$$J(\theta) = \mathbb{E}_{\tau \sim \pi_\theta}[r(\phi(\tau))] = \sum_{o \in \mathcal{O}} p_\theta(o)\, r(o). \quad (1)$$

Crucially, reward is always weighted by the policy's own outcome probability.

To isolate the effect of the objective, we consider a minimal abstraction: an outcome-selection bandit. At each step, the policy samples an outcome $o \sim p \in \Delta^{|\mathcal{O}|-1}$ and receives reward $r(o)$. This removes state dynamics, exploration issues, and credit assignment. Any instability observed here is therefore intrinsic to the objective in Eq. (1). We parameterize $p$ via logits $z$ with $p = \text{softmax}(z)$ and analyze continuous-time gradient ascent (gradient flow) on $J$.

**Theorem 3.1.** *Under gradient flow on $J(z) = \sum_o p_o(z)\, r(o)$ with a softmax parameterization, define the (on-policy) advantage of outcome $o$ at training time $t$ as $a_t(o) = r(o) - \mathbb{E}_{o' \sim p_t}[r(o')]$*

*Then for any outcomes $i, j$ with $p_i(t), p_j(t) > 0$,*

$$\frac{d}{dt} \log \frac{p_i(t)}{p_j(t)} = p_i(t)\, a_i(t) \;-\; p_j(t)\, a_j(t). \quad (2)$$

Equation (2) holds independently of exploration, sampling noise, entropy bonuses, or KL regularization, as long as the objective remains expected return.

## 3.1. Interpretation: a simple positive feedback loop

Equation (2) shows that probability ratios are driven by a very simple quantity. For each outcome $o$, define its update signal as: $g_t(o) = p_o(t)\, a_t(o)$.

Then the dynamics reduce to

$$\frac{d}{dt} \log \frac{p_i(t)}{p_j(t)} \;=\; g_t(i) \;-\; g_t(j).$$

Now consider what happens step by step.

**Step 1: one outcome is slightly ahead.** At any time $t$, there will generically be at least one outcome $i$ whose update signal $g_t(i) = p_i(t)a_t(i)$ is larger than that of all other outcomes $j$. Exact ties require perfectly equal probabilities and advantages and are unstable; in practice they are broken immediately by random initialization or sampling noise.

**Step 2: log-ratios start increasing.** If for some outcome $i$ we have: $g_t(i) > g_t(j)$ for all $j \neq i$. Then for every other outcome $j$, $\frac{d}{dt} \log \frac{p_i(t)}{p_j(t)} > 0$. This means that the ratio $log \frac{p_i(t)}{p_j(t)}$ increases linearly over time. Linear growth of $\log \frac{p_i(t)}{p_j(t)}$ implies exponential growth of $\frac{p_i(t)}{p_j(t)}$. Thus $p_i(t)$ increases exponentially relative to every other $p_j(t)$.

**Step 3: positive feedback strengthens the winner.** As $p_i(t)$ grows, its signal $g_t(i) = p_i(t)a_t(i)$ also grows, even if the advantage $a_t(i)$ stays constant or is equal to that of other outcomes. At the same time, $p_j(t)$ shrinks for all $j \neq i$, making $g_t(j)$ smaller. This further increases the gap $g_t(i) - g_t(j)$, which increases the growth rate of $\log \frac{p_i(t)}{p_j(t)}$ even more.

**Step 4: a point of no return.** This creates a positive feedback loop: $g_t(i)$ slightly larger $\Rightarrow$ $p_i$ grows faster $\Rightarrow$ $g_t(i) = p_i a_i$ becomes even larger $\Rightarrow$ $p_i$ grows even faster. Once this loop starts, the dynamics move irreversibly toward $p_i(t) = 1$. There is no mechanism in the expected-return objective that can restore balance.

This argument does not depend on rewards being unequal. Even when rewards (and thus advantages) are equal, the initial probability multiplier $p_o(t)$ ensures that the slightly more likely outcome receives a larger update. Because deep RL policies are initialized randomly or with heuristics, such small asymmetries are unavoidable. Therefore, for softmax policies over terminal outcomes, optimizing expected return inevitably leads to outcome-level mode collapse. Which outcome wins is determined by the initial values of $p_o(t)a_t(o)$, but collapse itself is unavoidable. Preventing this behavior requires changing the objective so that probability does not multiply its own update.

(proof for this section is in Appendix A)

# 4. Our Method: Inverse Probability Scaling

Our objective is to learn a policy such that for any outcome $i$, $\pi_i \propto r_i$. We have seen that the objective of maximizing expected return cannot converge to such a policy over outcomes. We propose a Inverse Probability Scaling (IPS) of rewards as a principle to achieve our objective.

## 4.1. The scaled objective

The inverse-scaled terminal reward

$$\tilde{r}_\theta(o) := \frac{r(o)}{p_\theta(o)} \quad where \quad p_\theta(o) = \Pr_{\tau \sim \pi_\theta}[\phi(\tau) = o].$$

(3)

The corresponding IPS-RL objective is

$$J_{\mathrm{IPS}}(\theta) = \mathbb{E}_{o \sim p_\pi}[\tilde{r}(o)] = \mathbb{E}_{o \sim p_\pi}\left[\frac{r(o)}{p_\theta(o)}\right].$$

(4)

where $p_\theta(o)$ is a scalar and gradients don't flow through it. IPS fundamentally changes how terminal outcomes are credited in the learning signal.

## 4.2. Stationary Solution: reward-proportional outcomes

To isolate the effect of the objective, consider the outcome-selection bandit as in Section 3, with logits $z \in \mathbb{R}^K$ and $p = \mathrm{softmax}(z)$. Specializing the derivation in

**Theorem 4.1.** *Under gradient flow on* $J_{\mathrm{IPS}}(z) = \mathbb{E}_{o \sim p(z)}\left[\frac{r(o)}{p_o(z)}\right]$ *with* $p = \mathrm{softmax}(z)$ *and outcome rewards* $\{r(i)\}_{i=1}^K$, *during the training the logits evolve as*

$$\frac{dz_i(t)}{dt} = r(i) - p_i(t) \sum_{k=1}^K r(k).$$

(5)

*Equivalently, for any $i, j$ with $p_i(t), p_j(t) > 0$,*

$$\frac{d}{dt} \log \frac{p_i(t)}{p_j(t)} = \big(r(i) - r(j)\big) - \big(p_i(t) - p_j(t)\big) \sum_{k=1}^K r(k).$$

(6)

The IPS dynamics admit a simple stationary solution with a direct interpretation.

**Corollary 4.2** (Reward-proportional stationary distribution)**.** *Assume $\sum_{k=1}^K r(k) > 0$ and $r(i) \geq 0$ for all $i$. Then the distribution*

$$p_i^\star = \frac{r(i)}{\sum_{k=1}^K r(k)}$$

(7)

*is stationary for* (5) *and* (6)
*In particular, $p^\star$ satisfies $r(i) - p_i^\star \sum_k r(k) = 0$, for all outcome $i$.*

**Interpretation** Expected-return optimization amplifies popularity because the learning signal for an outcome is multiplied by its current probability. In contrast, IPS cancels this

multiplier: at stationarity point solution of our IPS objective, probability mass is allocated *proportionally to reward*, preventing outcome-level mode collapse in multimodal settings. (proof for this section is in Appendix A)

## 4.3. With KL regularization: interpolation instead of collapse prevention

Most practical policy optimization methods include KL regularization to a reference policy $\pi_{\mathrm{ref}}$, either to encourage stochasticity (entropy regularization) or to preserve desirable behaviors during fine-tuning. Under expected-return optimization, KL regularization is often relied upon to *counteract outcome-level collapse*, with the reference policy acting as a source of randomness or as an anchor to a pretrained distribution. However, as shown in section 3, the collapse pressure originates from the expected-return objective itself; KL regularization merely opposes this pressure and must be carefully tuned to prevent degeneracy. Under inverse probability scaling, it is no longer required to prevent collapse. Instead, it serves its intended purpose: inducing stochastic exploration when $\pi_{\mathrm{ref}}$ is uniform distribution, or preserving proximity to a pretrained or preference-aligned policy during fine-tuning. Consider the KL-regularized IPS objective

$$\max_\pi \ \mathbb{E}_{o \sim p_\pi}\left[\frac{r(o)}{p_\pi(o)}\right] - \beta \, \mathrm{KL}(\pi \,\|\, \pi_{\mathrm{ref}}).$$

(8)

Here, the KL term no longer "fights collapse". It simply interpolates between (i) the reward-proportional behavior induced by IPS and (ii) adherence to the reference policy, with $\beta$ controlling the trade-off.

## 4.4. From principle to algorithm: IPS-GRPO

Inverse probability scaling is a general principle: it can be instantiated in any on-policy method by replacing terminal rewards with inverse-scaled rewards. We adopt GRPO as a concrete instantiation and refer to the resulting method as **IPS-GRPO**. We emphasize that the contribution is IPS; GRPO is one compatible choice of optimizer. Given a group of $G$ sampled trajectories $\{\tau_g\}_{g=1}^G$, let $o_g = \phi(\tau_g)$ be the terminal outcome. For a terminal outcome $o$ We estimate outcome probabilities by group frequency $\hat{p}(o)$ and apply a variance-stabilized scaling

$$\tilde{r}(o_g) = \frac{r(o_g)}{\max(\hat{p}(o_g), \epsilon)},$$

(9)

where $\epsilon > 0$ prevents extreme weights when $\hat{p}(o)$ is small.

# 5. Experiments

We evaluate IPS-GRPO in a range of outcome-multimodal settings to assess whether correcting outcome-frequency

**Algorithm 1** IPS-GRPO

---

**Require:** Policy $\pi_\theta$, terminal reward function $r(o)$, group size $G$, clipping $\epsilon$

1: **for** each update **do**
2:     Sample $G$ trajectories $\{\tau_g\}_{g=1}^{G} \sim \pi_\theta$
3:     Extract outcomes $o_g = \phi(\tau_g)$ and rewards $r_g = r(o_g)$
4:     Compute empirical outcome frequencies $\hat{p}(o) = \frac{1}{G}\sum_{g=1}^{G} \mathbf{1}\{o_g = o\}$
5:     Scale rewards $\tilde{r}_g = r_g / \max(\hat{p}(o_g), \epsilon)$
6:     Run a standard GRPO update using $\tilde{r}_g$ (all other terms and steps remains unchanged)
7: **end for**

---

bias in terminal rewards mitigates outcome-level mode collapse and yields terminal-state distributions that better reflect the underlying reward structure. Our evaluations span three settings: (i) a controlled discrete grid-based task, (ii) post-training of large language models on the Hypospace benchmark (Chen et al., 2025), and (iii) post-training of chemical language models for molecular generation (Moret et al., 2023).

### 5.1. Hyper-Grid Task

**Task setup.** A discrete $n$-dimensional hyper-grid environment with a multimodal terminal reward landscape. Episodes begin at $x_0 = (0, \ldots, 0)$, and actions consist of incrementing a single coordinate by $1$ or terminating the episode to receive the terminal reward. Each episode terminates at a grid coordinate $x \in \{0, \ldots, H-1\}^n$, and the reward is defined as

$$R(x) = R_0 + R_1 \prod_{i=1}^{n} \mathbf{1}(|x_i| > 0.5)$$
$$+ R_2 \prod_{i=1}^{n} \mathbf{1}(0.6 < |x_i| < 0.8), \tag{10}$$

with $0 < R_0 \ll R_1 < R_2$. This construction yields $2^n$ symmetric high-reward modes together with an enclosing ring structure, producing a sharply multimodal reward landscape (Figure 1). We define the target distribution over terminal states as $p(o) \propto R(o)$ and aim to learn a policy $\pi$ whose induced terminal-state distribution $\pi(o)$ matches $p(o)$. Since the correct outcome distribution is explicitly known, we evaluate methods using the $\ell_1$ distance between the empirical terminal-outcome distribution and the target distribution. We compare IPS-GRPO against GRPO and FlowRL under identical architectures, optimization settings, and sampling budgets. All of them have entropy regularisation to boost exploration. Experiments are conducted with $R_0 = 0.1$, $R_1 = 0.5$, and $R_2 = 2.0$ for $n = 2$ and $n = 4$.
**Results.** Figure 1 shows learned terminal-outcome distributions for the $n = 2$ setting. GRPO and FlowRL concentrate

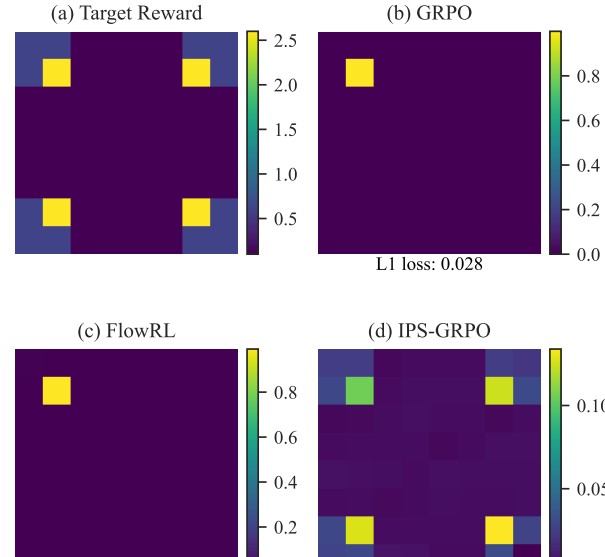

*Figure 1.* Learned terminal-state distributions on the 2D hyper-grid. (a) Target reward-induced distribution $p(x) \propto R(x)$. (b) GRPO and (c) FlowRL collapse onto a single high-reward mode despite multiple symmetric optima. (d) IPS-GRPO successfully recovers all modes and closely matches the target distribution. Reported values denote $\ell_1$ distance between the learned and target distributions.

| Method | $n = 2$ | $n = 4$ |
|---|---|---|
| GRPO | $2.8 \times 10^{-2}$ | $5.2 \times 10^{-4}$ |
| FlowRL | $2.8 \times 10^{-2}$ | $5.2 \times 10^{-4}$ |
| **IPS-GRPO** | $\mathbf{3.0 \times 10^{-3}}$ | $\mathbf{3.7 \times 10^{-4}}$ |

*Table 1.* $\ell_1$ distance between the learned terminal-state distribution and the target distribution on the hyper-grid domain. Lower is better.

the majority of probability mass on a single high-reward mode, exhibiting severe outcome-level mode collapse. In contrast, IPS-GRPO distributes probability mass across all symmetric modes and closely matches the target distribution. Table 1 reports quantitative results.

### 5.2. HypoSpace Benchmark

*HypoSpace* is a benchmark for LLMs which has set-valued inference problems with multiple admissible outcomes (Chen et al., 2025). This task is suitable to study the ability of LLMs to generate uniquely valid and different solutions.
**Task setup.** HypoSpace spans in structural, spatial, and symbolic reasoning: (i) causal inference, where agents generate causal graphs consistent with intervention data; (ii) 3D voxel reconstruction, where agents reconstruct gravity-consistent structures from top-down projections; and (iii) Boolean/DNA interaction, where agents propose Boolean programs consistent with observed phenotypes. Across all tasks, the full set of admissible outcomes is finite, allowing exact measurement of coverage.

**Baselines and metric.** We compare IPS-GRPO against GRPO and FlowRL under identical sampling budgets and reward definitions and KL regularization with the pretrained LLM model. We measure performance using *Recovery Rate* (RR), defined as the fraction of admissible outcomes recovered by samples from the learned policy of fine-tuned LLM. To characterize exploration dynamics, we additionally report the average number of distinct outcomes recovered as a function of the number of samples drawn.

**Results.** Figure 2 shows outcome recovery as a function of sampling budget across all three domains. GRPO and FlowRL exhibit early saturation: after a limited number of samples, additional draws yield few new outcomes, indicating premature concentration on a small subset of solutions. In contrast, IPS-GRPO continues to recover new outcomes as sampling increases, approaching wider coverage under identical budgets. Table 2 reports final recovery rates. Across all domains, IPS-GRPO achieves substantially higher recovery rate than both baselines. (Experimentation details are given in Appendix B.1)

### 5.3. Drug Discovery with Chemical Language Models

We finally apply IPS-GRPO to drug-discovery where diversity and quality of outcome is crucial. Chemical Language Models (CLMs) have seen success in discovering molecules in clinical trials. We adapt two realistic reward functions from (Guo et al., 2025): SYNTH and ALL-AMIDE that jointly reward binding potency and Synthesizability. The core CLM optimization problem is also a reward seeking RL problem with regularization: maximize rewards while staying close to the pretrained "prior" model to ensure chemical validity. Unlike traditional RL setting, CLMs are evaluated based on their ability to generate unique molecules given a fixed number of reward function evaluations. Which makes generating uniquely different molecules an essential quality for performance of CLMs. The REINVENT method (Olivecrona et al., 2017; Guo & Schwaller, 2024) is state-of-the-art RL based method on standard benchmarks (Gao et al., 2022). We apply IPS-GRPO as a policy learning algorithm as a replacement in their method.

Table 3 shows IPS-GRPO consistently results in higher Yield (number of unique high reward molecules discovered), and lower OB100 (number of samples needed to collect 100 such high reward molecules). Threshold represents the reward, such that the molecules accepted were above this threshold reward. where rewards is between 0 and 1. Throughout this experiment We see that IPS-GRPO outperforms the existing baseline of REINVENT. (Experimentation details are given in Appendix B.2)

### 5.4. Ablation: Group Size and Probability Clipping

IPS-GRPO relies on an empirical estimate of outcome probabilities computed within each sampled group. Two hy-

perparameters directly control the quality and stability of this estimate: the group size $G$ and the probability clipping threshold $\epsilon$ (Eq. 9). We study their effects on performance using the HypoSpace causal inference task, reporting Recovery Rate (RR).

We vary the group size $G \in \{4, 8, 16, 32, 64\}$ and the clipping threshold $\epsilon \in \{0.01, 0.1, 0.2\}$. Increasing $G$ improves the accuracy of the empirical probability estimate. However, for large groups, rare outcomes can receive very small estimated probabilities, leading to large inverse probabilities and unstable updates.

Probability clipping stabilizes training by bounding smaller probabilities. Larger $\epsilon$ values reduce variance and improve robustness when outcome probabilities are small, but aggressive clipping biases the probability estimate and weakens the intended inverse-probability correction. This can limit performance when group sizes are small and probability estimates are already coarse.

The ablation reveals a clear bias–variance trade-off: larger groups improve estimation accuracy but increase sensitivity to small probabilities, while clipping improves stability at the cost of correction fidelity. In practice, moderate group sizes combined with mild clipping provide the best balance (Table 4)

## 6. Why and How IPS-GRPO Works

Standard RL objectives conflate an outcome's quality with its reachability, leading to inevitable collapse. We use a controlled grid experiment with two terminal outcomes of equal reward but unequal path multiplicity with Outcome A (35 paths) and Outcome B (7 paths) to illustrate this pathology and our correction. (Figure 3)

Despite identical rewards, GRPO with entropy regularization rapidly collapses onto the high-multiplicity Outcome A. This is not a failure of discovery; both modes are found early in training. Instead, the collapse is driven by a structural bias: because each trajectory contributes proportionally to its sampling probability, Outcome A accumulates a larger global gradient signal. The trained policy force field reveals that GRPO vectors across most of the state space point toward A, creating a "rich-get-richer" loop that drives the frequency of B to zero.(Figure 3d 3c)

IPS-GRPO with entropy regularization corrects this by scaling rewards by the inverse empirical outcome frequency. This cancels the popularity multiplier, effectively decoupling reachability from preference. In the policy force field, IPS-GRPO maintains balanced magnitudes toward both outcomes, ensuring they compete solely on reward.(Figure 3b)

The qualitative difference in training dynamics is stark: GRPO reaches a winner-take-all state where nearly 100% of

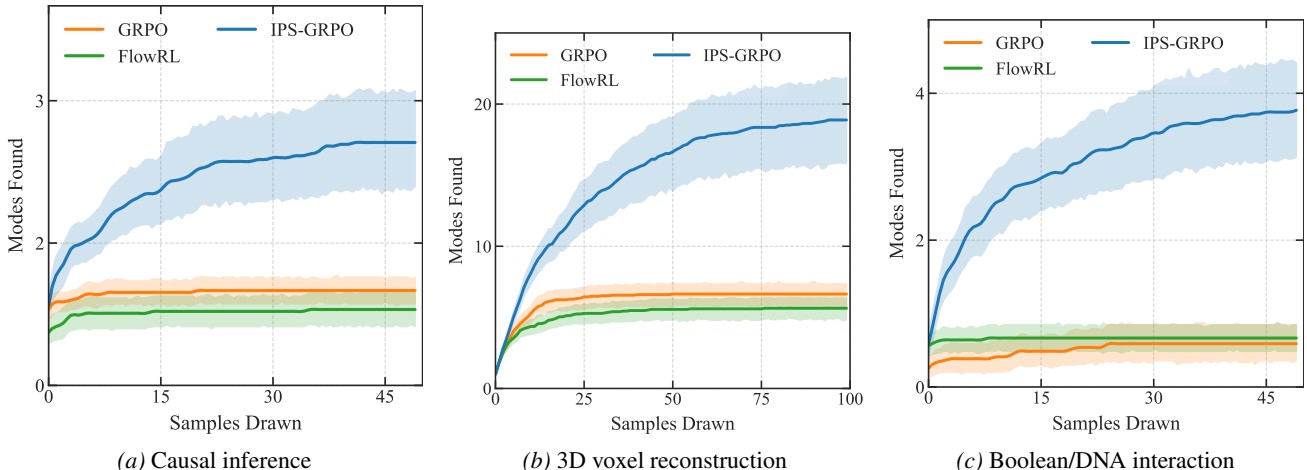

*(a)* Causal inference     *(b)* 3D voxel reconstruction     *(c)* Boolean/DNA interaction

*Figure 2.* Average number of distinct admissible modes recovered as a function of the number of samples drawn from the trained policy. Across all three HypoSpace domains, IPS-GRPO steadily discovers new modes with increased sampling, while GRPO and FlowRL saturate early, indicating outcome-level mode collapse.

| Method | Causal Inference | 3D Reconstruction | Boolean/DNA |
|---|---|---|---|
| GRPO | $16.02\% \pm 0.35\%$ | $31.61\% \pm 0.01\%$ | $9.50\% \pm 0.00\%$ |
| FlowRL | $12.82\% \pm 0.05\%$ | $26.85\% \pm 0.37\%$ | $10.74\% \pm 0.00\%$ |
| **IPS-GRPO** | $\mathbf{43.91\% \pm 1.39\%}$ | $\mathbf{90.00\% \pm 1.24\%}$ | $\mathbf{60.74\% \pm 0.78\%}$ |

*Table 2.* Final recovery rate (%) on HypoSpace tasks. Average over 5 random seeds. Higher is better.

| Threshold | Algorithm | Yield ↑ | OB100 ↓ |
|---|---|---|---|
| | | **SYNTH** | |
| 0.80 | REINVENT | $5695 \pm 173$ | $605 \pm 41$ |
| 0.80 | **IPS-GRPO** | $\mathbf{7371 \pm 212}$ | $\mathbf{554 \pm 34}$ |
| 0.85 | REINVENT | $418 \pm 78$ | $2934 \pm 68$ |
| 0.85 | **IPS-GRPO** | $\mathbf{1580 \pm 58}$ | $\mathbf{1178 \pm 43}$ |
| | | **ALL-AMIDE** | |
| 0.80 | REINVENT | $1 \pm 1$ | $> 5000$ |
| 0.80 | **IPS-GRPO** | $\mathbf{1478 \pm 102}$ | $\mathbf{773 \pm 51}$ |
| 0.85 | REINVENT | $0$ | $> 5000$ |
| 0.85 | **IPS-GRPO** | $\mathbf{20 \pm 12}$ | $> 5000$ |

*Table 3.* Molecular optimization results. Yield denotes the number of unique molecules exceeding the reward threshold; OB100 denotes oracle calls required to find 100 such molecules. The oracle budget for SYNTH is 10,000 and for ALL-AMIDE is 5,000. Average over 3 random seeds

| Group size $G$ | $\epsilon = 0.01$ | $\epsilon = 0.1$ | $\epsilon = 0.2$ |
|---|---|---|---|
| 4 | $17.63 \pm 0.33$ | $17.63 \pm 0.33$ | $17.63 \pm 0.33$ |
| 8 | $27.50 \pm 0.67$ | $27.50 \pm 0.67$ | $41.99 \pm 0.80$ |
| 16 | $41.02 \pm 0.60$ | $33.05 \pm 0.62$ | $\mathbf{43.91 \pm 1.39}$ |
| 32 | $32.13 \pm 0.42$ | $35.27 \pm 1.01$ | $25.62 \pm 1.65$ |
| 64 | $36.52 \pm 0.32$ | $39.12 \pm 1.33$ | $42.19 \pm 0.14$ |

*Table 4.* Recovery rate (%) on HypoSpace causal inference for different group sizes $G$ and probability clipping thresholds $\epsilon$. Average over 5 random seeds. Higher recovery rate is better.

samples are A. IPS-GRPO maintains stable, comparable frequencies for both A and B throughout training. (Figure 3c)

At the end of training, IPS-GRPO achieves a sampling density that closely matches the target reward distribution. While the policy still collapses to a subset of representative paths within each mode, it preserves essential diversity at the outcome level, confirming that IPS shifts the stationary solution of the objective itself.

## 7. Discussion and Conclusion

Our work shows outcome-level mode collapse as a structural consequence of the expected-return objective, rather than a failure of exploration, regularization, or optimization. We show that maximizing expected return inherently creates a positive feedback loop in which one dominant outcome receives larger updates and grows more dominant over training. This mechanism operates even under idealized conditions with perfect exploration and no optimization noise, implying that collapse is inevitable as long as expected return remains the objective. From this perspective, entropy regularization, KL constraints, and exploration bonuses do not eliminate collapse; they only oppose it and therefore soften its effects. This explains why diversity often degrades late in training in domains such as LLM post-training, despite careful tuning. Inverse Probability Scaling (IPS) addresses this issue at its source by removing probability amplification from the learning signal. By rescaling

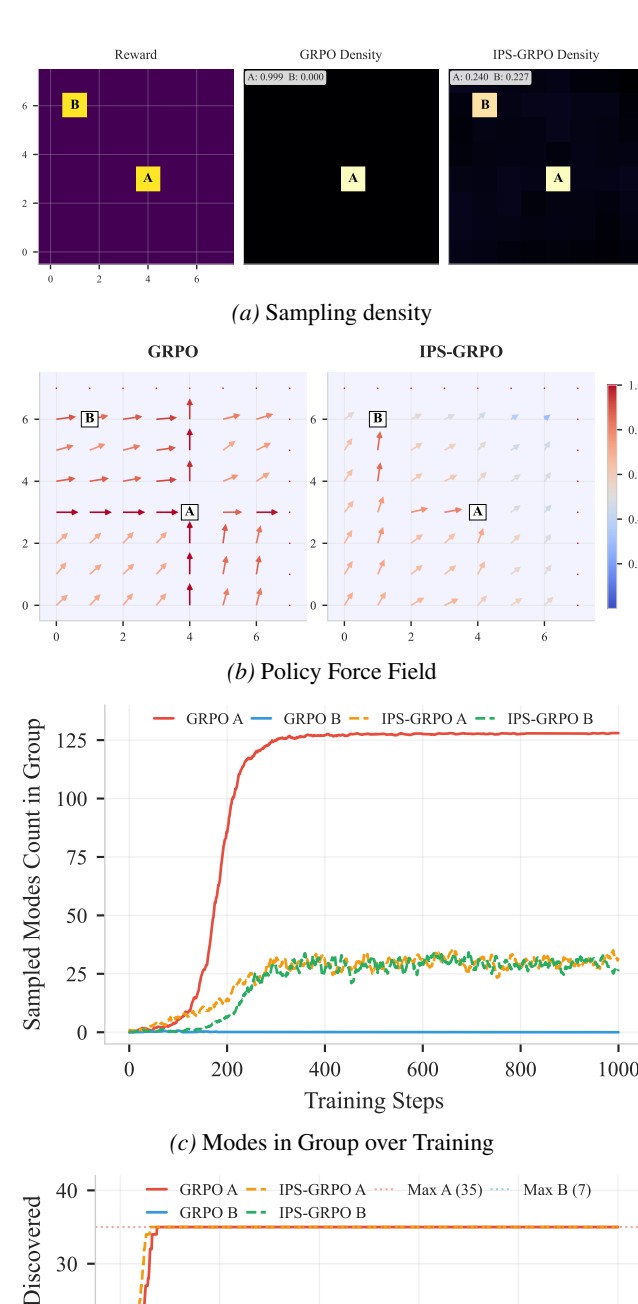

*(a)* Sampling density

*(b)* Policy Force Field

*(c)* Modes in Group over Training

*(d)* Path exploration during training

*Figure 3.* Study of IPS-GRPO behaviour against GRPO

*Table 5.* Summary of the paper's main claims and where they are supported.

| Claim | Evidence in This Work |
| --- | --- |
| Expected-return optimization can induce outcome-level mode collapse under idealized learning dynamics. | Section 3; log-ratio dynamics under gradient flow (Theorem 3.1, Equation (2)). |
| Probability-weighted updates contribute to collapse, including in settings with equal rewards. | Section 3.1; controlled equal-reward experiment (Figure 3). |
| Inverse Probability Scaling mitigates outcome-frequency amplification at the objective level. | IPS objective and dynamics in Section 4 (Theorems 4.1 and 4.2). |
| IPS-GRPO integrates IPS into standard policy-gradient pipelines with minimal modification. | Algorithm 1; hyper-grid recovery results (Figure 1 and Table 1). |
| IPS-GRPO yields improved outcome coverage in multimodal reasoning and discovery tasks. | HypoSpace benchmarks (Figure 2 and Table 2); molecular generation (Table 3). |

terminal rewards by the inverse outcome probability, IPS decouples outcome quality from outcome probability, yielding reward-proportional outcome distributions. IPS-GRPO demonstrates that this correction can be implemented within standard group based policy-gradient updates, achieving diversity preservation without any auxiliary models or explicit distribution-matching constraints.

### 7.1. Limitations and Future Directions

IPS relies on empirical estimates of outcome probabilities computed from finite groups of samples. In large or continuous outcome spaces, these estimates can be noisy, requiring probability clipping that introduces a bias–variance trade-off. Importantly, IPS is not directly applicable to RL methods that learn policies through Actor-Critic dynamic, such as PPO with learned critics, where updates depend on value estimates rather than explicit outcome frequencies. Extending IPS-like corrections to value-based or actor–critic methods remains an open problem. Finally, our analysis focuses on terminal rewards; extending IPS to dense-reward or continuous-outcome settings poses additional challenges.

### 7.2. Conclusion

Overall, this work shows that preserving diversity in multimodal RL requires rethinking the objective being optimized. By identifying outcome-level mode collapse as an objective pathology and introducing a minimal correction that removes outcome-frequency amplification, we demonstrate that stable and diverse outcome distributions can be achieved without abandoning standard RL pipelines. This suggests that objective design, not just optimization heuristics, is central to reliable multimodal reinforcement learning.

## Impact Statement

This paper studies outcome-level mode collapse in reinforcement learning and proposes Inverse Probability Scaling

(IPS) as a simple correction for preserving coverage over multiple high-reward terminal outcomes. The main positive impact of this work is to improve reliability in domains where diversity among valid solutions is important, such as language-model post-training, scientific discovery, program synthesis, and molecular generation. By reducing premature concentration on a small subset of outcomes, IPS may help RL systems better represent the structure of multimodal reward landscapes.

At the same time, methods that improve coverage of high-reward outcomes require careful reward design and domain-specific safeguards. In molecular generation, broader exploration should be paired with chemical-validity constraints, toxicity filters, and application-specific safety checks. In language-model post-training, preserving diversity should be combined with factuality, harmlessness, and preference-alignment constraints, since increased diversity can also preserve undesirable or low-quality modes if these are rewarded or insufficiently penalized.

The method does not introduce new data-collection procedures, human-subject experiments, surveillance capabilities, or deployment-specific risks beyond those already associated with reinforcement learning for generative models. Its impact depends primarily on the reward function, the outcome definition, and the safety constraints used in the downstream application.

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

# Appendix

# A. Mathematical Discussion

### A.1. PROOF OF THEOREM 3.1

*Proof.* Let $\mathcal{O} = \{1, \ldots, K\}$.

At continuous time $t$, the parameter is a logit vector $z(t) \in \mathbb{R}^K$ and the policy is the softmax distribution

$$p(t) = softmax(z(t)), \qquad p_i(t) = \frac{e^{z_i(t)}}{\sum_{k=1}^{K} e^{z_k(t)}}. \tag{1}$$

An outcome is sampled on-policy:

$$o_t \sim p(t).$$

A scalar reward $R_t$ is observed with $\mathbb{E}[|R_t|] < \infty$ and conditional mean

$$\mathbb{E}[R_t \mid o_t = i] = r_i,$$

where $r_i \in \mathbb{R}$ is fixed (independent of $z$). Define the expected-return objective

$$J(z) = \mathbb{E}_{o \sim softmax(z)}[r_o] = \sum_{k=1}^{K} p_k \, r_k, \tag{2}$$

Along the trajectory $z(t)$, define the on-policy mean reward and advantage:

$$\bar{r}(t) = \sum_{k=1}^{K} p_k(t) \, r_k, \qquad a_i(t) = r_i - \bar{r}(t).$$

Step 1 (sampling-aware gradient accumulation). Consider the score-function estimator

$$g(t) := R_t \, \nabla_z \log p_{o_t}(z(t)).$$

Because sampling is on-policy, the expected accumulated contribution of outcome $k$ is weighted by its sampling probability $p_k(t)$:

$$\mathbb{E}[g(t) \mid z(t)] = \sum_{k=1}^{K} \Pr[o_t = k \mid z(t)] \, \mathbb{E}[R_t \mid o_t = k] \, \nabla_z \log p_k(z(t))$$

$$= \sum_{k=1}^{K} p_k(t) \, r_k \, \nabla_z \log p_k(z(t)). \tag{A.1}$$

Step 2 (softmax log-derivative identity). For all $i, k \in \{1, \ldots, K\}$,

$$\frac{\partial \log p_k(z)}{\partial z_i} = \mathbf{1}\{i = k\} - p_i(z). \tag{A.2}$$

Step 3 (compute $\nabla_z J$ explicitly). Using (A.1) and (A.2), for each coordinate $i$,

$$\frac{\partial J(z(t))}{\partial z_i} = \sum_{k=1}^{K} p_k(t) \, r_k \, \frac{\partial \log p_k(z(t))}{\partial z_i}$$

$$= \sum_{k=1}^{K} p_k(t) \, r_k \, (\mathbf{1}\{i = k\} - p_i(t))$$

$$= p_i(t) \, r_i - p_i(t) \sum_{k=1}^{K} p_k(t) \, r_k$$

$$= p_i(t) \, (r_i - \bar{r}(t))$$

$$= p_i(t) \, a_i(t). \tag{A.3}$$

Thus, under gradient flow $\dot{z}(t) = \nabla_z J(z(t))$,

$$\boxed{\dot{z}_i(t) = p_i(t)\, a_i(t).}$$

(3)

The factor $p_i(t)$ is precisely the sampling-frequency weight: outcome $i$ is sampled with probability $p_i(t)$, so its update is accumulated with that frequency (Step 1).

Step 4 (log-ratio dynamics). For softmax, $\log p_i(t) = z_i(t) - \log\left(\sum_{\ell=1}^{K} e^{z_\ell(t)}\right)$, hence for any $i, j$,

$$\frac{d}{dt} \log \frac{p_i(t)}{p_j(t)} = \dot{z}_i(t) - \dot{z}_j(t).$$

Substituting (3) yields

$$\boxed{\frac{d}{dt} \log \frac{p_i(t)}{p_j(t)} = p_i(t)\, a_i(t) - p_j(t)\, a_j(t).}$$

(4)

This proves Theorem 3.1. □

## A.2. PROOF FOR THEOREM 4.1

*Proof.* we use a stop-gradient copy of the current probabilities. At time $t$, define

$$p(t) = softmax(z(t)), \qquad \bar{p}_i(t) := stopgrad(p_i(t)),$$

so $\bar{p}(t)$ is treated as constant with respect to $z$ when differentiating at time $t$.

Define IPS scaling and the IPS objective:

$$\boxed{\tilde{r}(o) = \frac{r_o}{\bar{p}_o(t)}.}$$

(5)

$$\boxed{J_{\text{IPS}}(z(t)) = \mathbb{E}_{o\sim p(t)}\left[\frac{r_o}{\bar{p}_o(t)}\right] = \sum_{k=1}^{K} p_k(t)\frac{r_k}{\bar{p}_k(t)}.}$$

(6)

Step 1 (sampling-aware gradient accumulation). With $o_t \sim p(t)$, the score-function estimator is

$$g_{\text{IPS}}(t) = \frac{r_{o_t}}{\bar{p}_{o_t}(t)}\, \nabla_z \log p_{o_t}(z(t)).$$

Taking conditional expectation and using on-policy sampling:

$$\mathbb{E}[g_{\text{IPS}}(t) \mid z(t)] = \sum_{k=1}^{K} p_k(t)\frac{r_k}{\bar{p}_k(t)}\, \nabla_z \log p_k(z(t)) = \nabla_z J_{\text{IPS}}(z(t)),$$

(A.7)

since $\bar{p}_k(t)$ is constant w.r.t. $z$ at time $t$.

Step 2 (compute the gradient coordinates). Let $w_k(t) = r_k/\bar{p}_k(t)$, so $J_{\text{IPS}}(z(t)) = \sum_k p_k(t)w_k(t)$. Using $\partial p_k/\partial z_i = p_k(\mathbf{1}\{i = k\} - p_i)$,

$$\begin{aligned}
\frac{\partial J_{\text{IPS}}(z(t))}{\partial z_i} &= \sum_{k=1}^{K} w_k(t)\frac{\partial p_k(z(t))}{\partial z_i} \\
&= \sum_{k=1}^{K} w_k(t)\, p_k(t)\, (\mathbf{1}\{i = k\} - p_i(t)) \\
&= p_i(t)\, w_i(t) - p_i(t)\sum_{k=1}^{K} p_k(t)\, w_k(t) \\
&= p_i(t)\, w_i(t) - p_i(t)\, J_{\text{IPS}}(z(t)).
\end{aligned}$$

(A.8)

Because $\bar{p}_k(t) = stopgrad(p_k(t))$ equals $p_k(t)$ numerically at time $t$,

$$w_i(t) = \frac{r_i}{p_i(t)}, \qquad J_{\text{IPS}}(z(t)) = \sum_{k=1}^{K} p_k(t)\frac{r_k}{p_k(t)} = \sum_{k=1}^{K} r_k =: R.$$

Substituting into (A.8) yields the IPS gradient flow

$$\boxed{\dot{z}_i(t) = r_i - R\, p_i(t).} \tag{7}$$

Step 3 (log-ratio dynamics). As in Appendix A.1, $\frac{d}{dt}\log\frac{p_i}{p_j} = \dot{z}_i - \dot{z}_j$, so by (7),

$$\boxed{\frac{d}{dt}\log\frac{p_i(t)}{p_j(t)} = (r_i - r_j) - (p_i(t) - p_j(t))R.} \tag{8}$$

This is Theorem 4.1. In particular, if $r_i = r_j$ then the term $-(p_i - p_j)R$ is a restoring force toward $p_i = p_j$.

Stationary point. Assume $r_i \geq 0$ and $R = \sum_k r_k > 0$. Define

$$\boxed{p_i^\star = \frac{r_i}{\sum_{k=1}^{K} r_k} = \frac{r_i}{R}.} \tag{9}$$

Then $\dot{z}_i(t) = r_i - Rp_i^\star = 0$ for all $i$, so $p^\star$ is stationary.

Global convergence (potential function). Define

$$\Psi(z) = \log\Big(\sum_{k=1}^{K} e^{z_k}\Big) - \sum_{k=1}^{K} p_k^\star z_k.$$

Then $\nabla_z \Psi(z) = p(z) - p^\star$, and by (7), $\dot{z}(t) = R(p^\star - p(t))$. Hence

$$\frac{d}{dt}\Psi(z(t)) = \langle p(t) - p^\star,\ R(p^\star - p(t))\rangle = -R\|p(t) - p^\star\|_2^2 \leq 0.$$

Therefore $\Psi(z(t))$ is nonincreasing and $\int_0^\infty \|p(t) - p^\star\|_2^2\, dt < \infty$, which implies $p(t) \to p^\star$ as $t \to \infty$. $\qquad\square$

# B. EXPERIMENT DETAILS

## B.1. HYPOSPACE

### CAUSAL DISCOVERY

**Dataset construction.** Datasets are synthetically generated by exhaustively enumerating all directed acyclic graphs (DAGs) under fixed structural constraints and grouping them by their intervention signatures. Each graph has 4 nodes $\{A, B, C, D\}$ with up to 6 directed edges. For each graph, observations are produced by perturbing a subset of nodes (3 perturbations per sample) and recording binary effects on all other nodes, determined by causal descendants. This yields a finite hypothesis set with exact ground truth coverage.

| | |
|---|---|
| Number of nodes | 4 |
| Max edges | 6 |
| Observations per sample | 3 |
| Total samples | 100 (50 train / 50 test) |

### 3D STRUCTURE RECONSTRUCTION

**Dataset construction.** Data are generated by exhaustively enumerating valid voxel structures consistent with gravity and a given top-down projection. Top views are created by selecting up to 3 occupied cells in a $3 \times 3$ grid. For each top view, all height assignments (1 to 3 layers) are enumerated and filtered to satisfy physics constraints: every occupied voxel at layer $h$ must be supported at layer $h-1$. Each top view corresponds to a finite set of valid 3D hypotheses.

| | |
|---|---|
| Grid size | $3 \times 3$ |
| Max visible blocks | 3 |
| Max height | 3 |
| Blocks fixed | No (1–3 blocks) |
| Total samples | 100 (50 train/50 test) |

## BOOLEAN EXPRESSION DISCOVERY

**Dataset construction.** Boolean datasets are generated via exhaustive enumeration of expressions over two variables $\{x, y\}$ with operators AND, OR, and NOT, up to depth 2. Expressions are canonically deduplicated using mechanistic equivalence rules (commutativity, idempotence, and associativity flattening). For each partial input–output observation set, all expressions consistent with the induced truth table are treated as valid hypotheses.

| | |
|---|---|
| Variables | $2\ (x, y)$ |
| Operators | AND, OR, NOT |
| Max expression depth | 2 |
| Total Samples | 78 (39 train / 39 test) |

## MODEL AND PARAMETER-EFFICIENT FINE-TUNING

**Base model.** All experiments use `Gemma-3 4B Instruct` as the underlying language model, with 4-bit NF4 quantization and bfloat16 computation.

**LoRA configuration.** A single LoRA setup is shared across all Hypospace tasks.

| | |
|---|---|
| LoRA rank $(r)$ | 16 |
| LoRA $\alpha$ | 32 |
| Target modules | $q, k, v, o$ projections |
| Dropout | 0.05 |

**Reward.** Each generated hypothesis receives a binary reward based on if it is valid. A response is first parsed into a structured hypothesis (graph, 3D structure, or Boolean expression). The hypothesis is validated against the ground truth set by checking structural equivalence using hash-based matching for graphs and structures, and mechanistic keys for Boolean expressions. A reward of 1.0 is assigned only if the hypothesis is valid which matches a ground truth solution. Invalid parses, incorrect hypotheses.

**Evaluation.** Models are evaluated via *Cumulative Recovery*, computed from parallel sampling.

Hypospace GitHub link [Hypospace](Hypospace)

### B.2. Drug Discovery with Chemical Language Models

**Multi-objective reward function.** For both the **SYNTH** and **ALL-AMIDE** tasks, molecule generation is guided by a multi-parameter optimization (MPO) reward composed of four components: docking score, drug-likeness (QED), hydrogen bond donor count (HBD), and synthesizability. These components are aggregated using a geometric mean with equal weighting,

$$R_{\text{final}} = \prod_{i=1}^{4} R_i^{\frac{1}{4}}. \tag{11}$$

**Docking score component.** Binding affinity is evaluated against the SARS-CoV-2 main protease using QuickVina2-GPU-2.1. Docking is performed on the crystal structure PDB 7UVU, prepared with PDBFixer. For each generated SMILES string, the following pipeline is applied: (i) conversion to an RDKit molecule, (ii) protonation at physiological pH, (iii) conformer generation using ETKDG, (iv) energy minimization with the Universal Force Field (UFF), (v) conversion to PDBQT format via OpenBabel, and (vi) docking with an exhaustiveness of 8000 using a $20 \times 20 \times 20$ Å$^3$ search box centered on the reference ligand.

The raw docking score $x$ (in kcal/mol, where more negative values indicate stronger binding) is transformed to a normalized

reward using a reverse sigmoid:

$$R_{\text{dock}} = \frac{1}{1 + 10^{k \cdot (x - (h+l)/2) \cdot \frac{10}{h-l}}}, \tag{12}$$

with parameters $l = -16$, $h = 0$, and $k = 0.15$, mapping favorable scores close to $-16$ kcal/mol to rewards near 1.

**Drug-likeness components.** Drug-likeness is quantified using the RDKit implementation of the Quantitative Estimate of Drug-likeness (QED), yielding values in $[0, 1]$ without further transformation.

**Hydrogen Bond Donor component** The hydrogen bond donor count is computed using RDKit and evaluated with a step function,

$$R_{\text{HBD}} = \begin{cases} 1.0, & 0 \leq \text{HBD} \leq 3, \\ 0.0, & \text{otherwise}, \end{cases} \tag{13}$$

enforcing a strict constraint.

**Synthesizability constraints.** Synthesizability is assessed using the Syntheseus retrosynthesis framework with the MEGAN molecular edit graph attention network. Commercially available eMolecules building blocks are used, with a time limit of 180 seconds per molecule.

For the **SYNTH** task, the synthesizability reward is binary and equals 1 if any valid synthesis route is found. For the **ALL-AMIDE** task, an additional constraint is imposed: all reactions in the synthesis route must belong to the *amide* or *acylation* reaction classes, as determined using Rxn-INSIGHT. The reward is set to 1 only if both synthesizability and reaction-class constraints are satisfied.

**Final reward definitions.** The complete reward functions are given by

$$R_{\text{SYNTH}} = \left( R_{\text{dock}} \cdot R_{\text{QED}} \cdot R_{\text{HBD}} \cdot R_{\text{synth}} \right)^{\frac{1}{4}}, \tag{14}$$

and

$$R_{\text{ALL-AMIDE}} = \left( R_{\text{dock}} \cdot R_{\text{QED}} \cdot R_{\text{HBD}} \cdot R_{\text{synth+amide}} \right)^{\frac{1}{4}}, \tag{15}$$

where $R_{\text{synth+amide}}$ is non-zero only if a valid synthesis route exists and all reactions satisfy the enforced amide-coupling constraints.

**Model and reinforcement learning setup.** We run our experiments on the top of the SATURN github repo and use the model trained by them. Saturn Github link[https://github.com/schwallergroup/saturn].

The chemical language model is based on a Mamba architecture with 12 layers, hidden dimension 256, RMS normalization, and approximately 5.27M parameters. The model is pretrained on ChEMBL 33 using maximum likelihood estimation with SMILES randomization for 18 epochs. Reinforcement learning fine-tuning follows the REINVENT framework with a batch size of 64, learning rate $10^{-4}$, KL coefficient 128, and an oracle budget of 10,000 evaluations. An identical Murcko scaffold diversity filter with bucket size 10 is applied, along with augmented memory and experience replay to promote structural diversity.

## C. RESULTS: 3D VOXELS

We illustrate outcome-level diversity using a single 3D voxel reconstruction instance with a fixed top-view projection over a 3×3 grid and a maximum height of three layers. The fine-tuned model is queried using a structured prompt that specifies the top view, enforces gravity and consistency constraints, and requires explicit layer-by-layer reconstruction. For this input, multiple distinct 3D voxel structures are possible, different height assignments satisfy the physics constraints while producing the same top-view projection under an OR operation across layers.

The prompt given to both models were

' ' '

```
### TASK DESCRIPTION
You are an expert 3D reconstruction engine.
You are given a Top View (2D projection) of a 3x3 grid.
Your goal is to reconstruct the 3D voxel structure
(layers of blocks) that matches this view.

Maximum allowed height: 3 layers.

### OBSERVATION
Top View (1=block visible, 0=empty):
0 1 0
0 0 0
0 1 1

### REQUIRED OUTPUT FORMAT
You must output the structure layer by layer, starting from the bottom (Layer 1).
Every layer must be explicitly labeled.
Do not output raw numbers without 'Layer X:' labels.

Use exactly this format:
BEGIN_STRUCTURE
Structure:
Layer 1:
0 0 0
0 1 0
0 0 0
Layer 2:
0 0 0
0 1 0
0 0 0
END_STRUCTURE

### CONSTRAINTS
1. Physics: A block at Layer N must have a block directly below it at Layer N-1.
2. Consistency: The 'OR' operation of all layers must exactly match the Top View.
3. Format: You must write 'Layer X:' before each grid.

Generate the structure now:

'''
```

Despite the small problem size and finite hypothesis space, standard expected-return optimization collapses onto a narrow subset of valid reconstructions. Figure 4

In contrast, IPS-based optimization consistently generates a diverse set of valid voxel structures for the same prompt. This demonstrates that IPS preserves diversity at the level of terminal outcomes, even in simple deterministic reasoning problems with fully enumerated solution sets. Figure 5

This example provides a concrete qualitative illustration of outcome-level mode collapse and its mitigation: IPS prevents early dominance of a single reconstruction and enables the model to represent the full space of valid 3D hypotheses induced by a fixed observation.

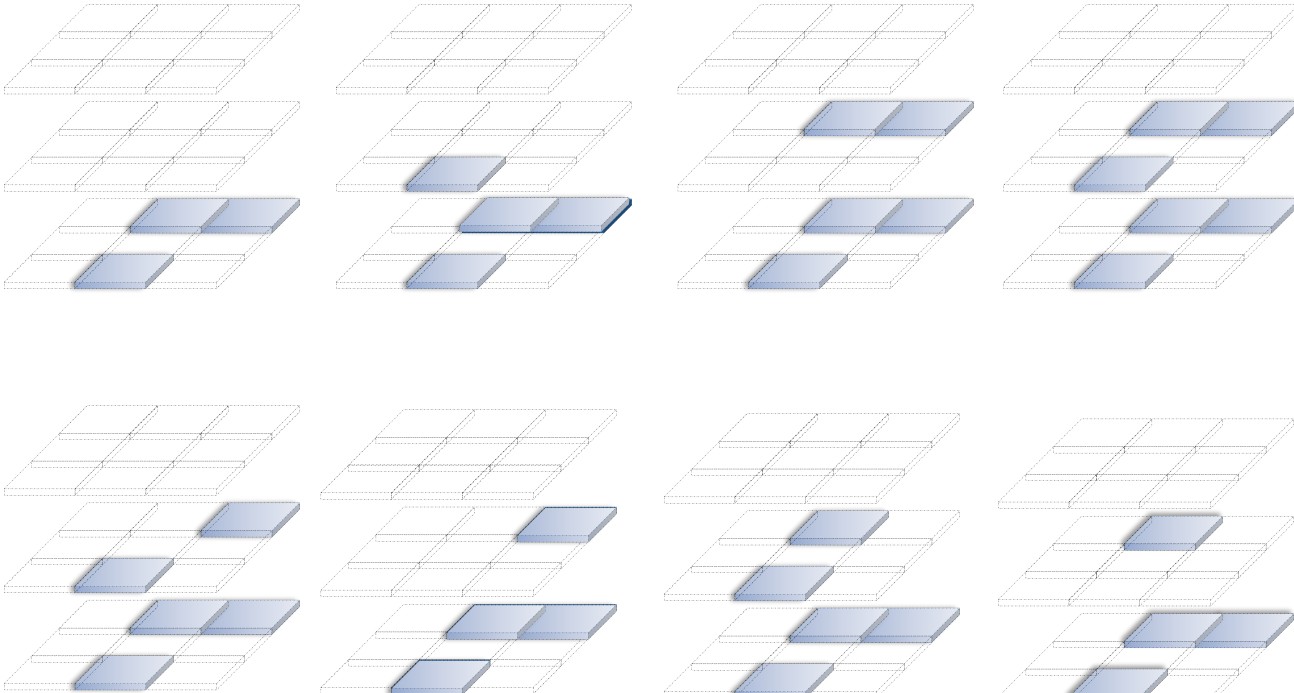

*Figure 4.* voxels generated by GRPO

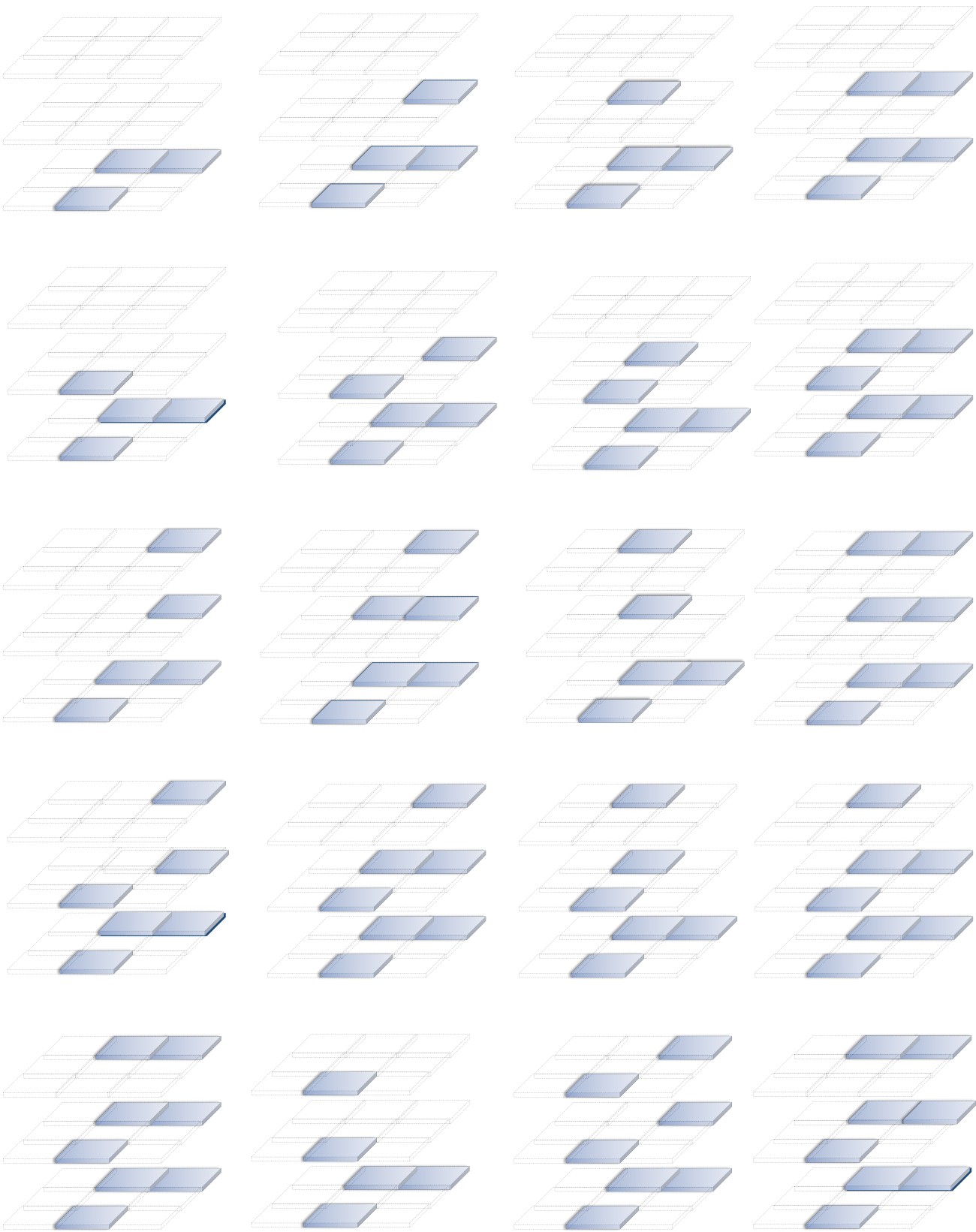

*Figure 5.* voxels generated by IPS-GRPO

