# OpenReview forum: "Expected Return Causes Outcome-Level Mode Collapse in Reinforcement Learning and How to Fix It with Inverse Probability Scaling"
_ICML.cc/2026/Conference — ICML 2026 regular_

### Official Review · Reviewer_1uhv · 2026-02-23

**Soundness:** 3
**Presentation:** 3
**Significance:** 3
**Originality:** 3
**Overall Recommendation:** 4
**Confidence:** 3

**Summary:**

This paper talks about outcome-level mode collapse in RL, especially when there are several good final good outcomes instead of just one best solution. The main point is that if we simply maximize expected return, the policy will often focus on only one outcome, even though other outcomes have very similar rewards. The authors argues this happens because of the objective itself. Since the gradient is multiplied by the current probability of each outcome, slightly more frequent outcomes get larger updates. Over time, this creates a positive feedback effect and the policy collapses to one mode. To address this, they propose Inverse Probability Scaling (IPS), which divides the reward by the outcome probability. This reduces the amplification effect and leads to a distribution that is closer to being proportional to reward. They apply this idea to GRPO (IPS-GRPO) and show improved diversity on grid tasks, reasoning benchmarks, and molecular generation.

In my view, the key contribution is highlighting that the collapse comes from objective design and proposing a simple fix that can be added to existing RL methods.

**Compliance With Llm Reviewing Policy:**

Affirmed.

**Key Questions For Authors:**

**1.	Scalability to large outcome spaces**

In large & continuous outcome spaces (e.g., LLM generation), how stable is the empirical probability estimation? If the estimation becomes very noisy, does IPS still work well? If the authors can show strong results in larger-scale settings, it would increase my confidence in the practical impact.

**2.	Compatibility with actor-critic methods**

The paper mentions that IPS is not directly applicable to actor-critic methods like PPO with learned critics. Do the authors have any idea or preliminary results on how to extend IPS to these common RL frameworks?

**3.	Effect on performance-quality trade-off**

Sometime improving diversity may hurt peak performance. I am curious if the author observe any trade-off between diversity and best-case reward in experiments? If IPS can keep both diversity and top performance at the same time, that would make the results more convincing.

**Limitations:**

Yes

**Strengths And Weaknesses:**

**Soundness**

I think the paper is generally technically sound. The theoretical analysis explains clearly why expected-return optimization can lead to outcome-level collapse. The math seems consistent, and the simplified setting helps to understand the core issue. The experiments covers different tasks, not only toy examples, which makes the results more convincing. However, I think more discussion about limitations and comparison with additional RL baseline would make it stronger.

**Presentation**

The paper is well organized and easy to follow. The motivation and main idea are understandable. Some theory parts are a bit dense and may be hard for readers without strong math background. Maybe adding more intuitive explanation will be better to understand.

**Significance**

The problem is relevant, especially for LLM fine-tuning and generative tasks where diversity is important. If the idea works well in larger settings, it may influence how people design RL objective in future.

**Originality**
The method is simple, but the insight that collapse comes from the objective itself is interesting. I think the originality mainly comes from this perspective and its practical implementations.

---

> ### Author Rebuttal · Authors · 2026-03-31
>
> ## Scalability to large outcome spaces
>  For very large and continuous space (e.g. LLM generation), $\hat{p}(o)$ calculation necessary for IPS algorithm would not be possible from empirical estimation and instead require a separate probability density estimator for $\hat{p}(o)$ estimation of the generated trajectory. We Hope to address this in the future work.
>
> ---
> ## Effect of Noise in Outcome Probability
> IPS-GRPO's ability to estimate outcome probabilities is sensitive to the value of the group size $G$ as the total outcome space grows. Experiments on a 2D grid task varying Horizon length $H$ and Group size $G$ reveal two key behaviors:
>
> - **Sensitivity to Estimation Noise:** As valid outcomes scale, small group sizes lead to poor distribution matching and failed mode discovery. To maintain stability in larger spaces, $G$ must scale proportionally.
> - **Bias from Variance Control:** In massive spaces with small $G$, most outcomes appear once, yielding small empirical probabilities. While the clipping threshold $\epsilon$ prevents reward explosion and controls variance, it introduces bias by weakening the IPS correction. This causes the algorithm to exhibit the early saturation typical of standard GRPO.
> - **Experimental Results**
> The values in the table are: KL Divergence (no of outcomes sampled)
> | $G$ ↓ | $H=8$ [64 outcomes] | $H=16$ [256 outcomes] | $H=32$ [1024 outcomes] | $H=64$ [4096 outcomes] |
> |--------|----------------------|------------------------|-------------------------|-------------------------|
> | 8 | 1.38 (21) | 1.98 (27) | 2.54 (100) | 2.66 (170) |
> | 16 | 0.56 (31) | 1.74 (29) | 2.51 (51) | 2.59 (167) |
> | 32 | 0.25 (43) | 1.73 (30) | 2.31 (76) | 2.63 (138) |
> | 64 | 0.19 (53) | 0.38 (134) | 1.78 (86) | 2.49 (133) |
> | 128 | 0.03 (64) | 0.36 (143) | 0.78 (281) | 2.31 (222) |
> | 256 | 0.07 (64) | 0.09 (254) | 0.44 (326) | 1.81 (269) |
> | 512 | 0.11 (64) | 0.04 (256) | 0.30 (354) | 1.14 (578) |
> | 1024 | 0.10 (64) | 0.09 (256) | 0.15 (786) | 1.03 (652) |
> | 2048 | 0.10 (64) | 0.09 (256) | 0.06 (1024) | 0.32 (1254) |
>
>
> ---
> ## Applying IPS to Other RL Algorithms
>
> Extending IPS to actor-critic methods like PPO is non-trivial as they rely on learned value or advantage estimates. Because IPS dynamically scales terminal rewards, a "moving-target" mismatch might be introduced between the critic's value estimates and the actor's update objective. Hence, we used GRPO, a critic-free method, as our testbed.
>
> To demonstrate IPS is not tied to GRPO, we applied it to REINFORCE. In this setting, we estimate the outcome probability $\hat{p}(o)$ using empirical frequencies within a rollout batch $\mathcal{B}$ of $N$ trajectories:
>
> $$\hat{p}(o) = \frac{1}{N} \sum_{i=1}^N \mathbb{1}\{ \phi(\tau_i) = o \}$$
>
> Applying the variance-stabilizing threshold $\epsilon$, the IPS-REINFORCE policy gradient estimator is:
>
> $$\nabla_\theta J_{IPS}(\theta) = \frac{1}{N} \sum_{n=1}^N \left[ \sum_{t=0}^{T-1} \nabla_\theta \log \pi_\theta(a_t | s_t) \right] \frac{r(o_n)}{\max(\hat{p}(o_n), \epsilon)}$$
>
> To validate this formulation, we evaluated IPS-REINFORCE against vanilla REINFORCE on the 2D hyper-grid environment extended to a horizon length of 8 (comprising 64 total distinct outcomes), with the following findings:
>
> | Algorithm | KL divergence | Outcomes sampled
> |-----------|----------|---------------|
> | Vanilla REINFORCE | 2.153550 | 1 / 64 |
> | IPS-REINFORCE (Ours) | 0.050460 | 64 / 64 |
>
> Vanilla REINFORCE suffers from severe outcome-level mode collapse, whereas IPS-REINFORCE successfully maintains coverage and accurately recovers the target distribution.
>
> ---
> ## Effect of performance-quality trade-off
>
> IPS-GRPO discovers more unique solutions without lowering the quality threshold. This is verified in three experimental settings, we will present CLM findings:
>  Quality here is enforced by a high reward threshold.  IPS-GRPO finds more molecules above this bar with fewer oracle calls, achieving a higher peak reward than REINVENT.
>
> | Metric / Reward Bracket | REINVENT (SYNTH) | IPS-GRPO (SYNTH) | REINVENT (ALL-AMIDE) | IPS-GRPO (ALL-AMIDE) |
> |:---|:---:|:---:|:---:|:---:|
> | Absolute Max Reward Achieved | 0.8526 | 0.8663 | 0.7223 | 0.8531 |
> | $[0.00,\ 0.80)$ | 90 (11.1%) | 56 (7.1%) | 487 (100.0%) | 99 (26.3%) |
> | $[0.80,\ 0.81)$ | 14 (1.7%) | 16 (2.0%) | 0 (0.0%) | 29 (7.7%) |
> | $[0.81,\ 0.82)$ | 84 (10.3%) | 63 (8.0%) | 0 (0.0%) | 28 (7.4%) |
> | $[0.82,\ 0.83)$ | 192 (23.6%) | 141 (18.0%) | 0 (0.0%) | 62 (16.4%) |
> | $[0.83,\ 0.84)$ | 305 (37.5%) | 181 (23.1%) | 0 (0.0%) | 45 (11.9%) |
> | $[0.84,\ 0.85)$ | 120 (14.8%) | 188 (23.9%) | 0 (0.0%) | 76 (20.2%) |
> | $[0.85,\ 0.86)$ | 8 (1.0%) | 122 (15.5%) | 0 (0.0%) | 38 (10.1%) |
> | $[0.86,\ 0.87)$ | 0 (0.0%) | 18 (2.3%) | 0 (0.0%) | 0 (0.0%) |

---

> > ### Author Rebuttal · Reviewer_1uhv · 2026-04-06
> >
> > The rebuttal addresses most of my concerns.
> >
> > The clarifications on scalability and the role of group size are helpful, and the extension beyond GRPO (e.g., REINFORCE) strengthens the case. The discussion on the diversity–performance trade-off is also convincing.
> >
> > Some limitations remain, particularly for actor–critic methods and large or continuous outcome spaces, but these are acknowledged and reasonable.
> >
> > Overall, the rebuttal improves the paper, and I keep my positive assessment.

---

### Official Review · Reviewer_pS98 · 2026-03-08

**Soundness:** 3
**Presentation:** 3
**Significance:** 3
**Originality:** 3
**Overall Recommendation:** 4
**Confidence:** 4

**Summary:**

This paper studies outcome-level mode collapse in reinforcement learning settings where multiple terminal outcomes can all be valid and high-reward, but standard expected-return optimization tends to concentrate probability mass on only a small subset of them. The paper argues that this phenomenon is not merely due to poor exploration, weak regularization, or optimization noise, but is instead a structural property of the expected-return objective itself. To make this case, the authors analyze an outcome-selection bandit abstraction and derive log-ratio dynamics showing that probability differences between outcomes are amplified over training, leading to collapse under idealized gradient-flow dynamics.

To address this issue, the paper proposes Inverse Probability Scaling (IPS), which rescales each terminal outcome’s reward by the inverse of its current outcome probability, with gradients stopped through that probability term. The empirical evaluation spans three settings.

**Compliance With Llm Reviewing Policy:**

Affirmed.

**Key Questions For Authors:**

1. How sensitive is IPS-GRPO to estimation noise in the outcome probabilities, especially as the number of valid outcomes grows?
2. Why is the method evaluated only as a GRPO-style modification, and what prevents a closer empirical test against critic-based policy optimization?
3. Can the authors reconcile the strong collapse narrative in the hyper-grid discussion with the small quantitative gaps in some reported results?

**Limitations:**

yes

**Strengths And Weaknesses:**

Strengths:
- The paper proposes Inverse Probability Scaling (IPS) and instantiates it as IPS-GRPO, a simple modification of GRPO that rescales terminal rewards by inverse estimated outcome frequency. The work combines a clean theoretical story with experiments on controlled hyper-grid environments, HypoSpace reasoning tasks, and molecular generation.
- The authors are not merely observing collapse empirically, but attempting to explain why it should arise even in idealized settings. The IPS correction is then derived in a way that directly targets that mechanism, and the paper argues that the corrected dynamics yield reward-proportional stationary behavior instead of winner-take-all concentration.

Weaknesses:
- The theoretical analysis is built on fairly idealized assumptions: terminal-reward dependence on finite outcomes, softmax logits, and continuous-time gradient flow. Those assumptions are acceptable for deriving insight, but they leave a gap between the proven statements and the broader claims one might want to make about modern RL pipelines.
- In the hyper-grid section, the text emphasizes severe collapse for GRPO and FlowRL and strong recovery by IPS-GRPO, but Table 1 includes an n=4 case where the reported ℓ1 distances for all methods are already very small, and the gap between baselines and IPS-GRPO becomes quite narrow. That does not invalidate the result, but it weakens the force of the narrative and should be explained more carefully. The figure/table presentation there feels under-polished.

---

> ### Author Rebuttal · Authors · 2026-03-31
>
> ## Assumptions in the theoretical analysis
> The reviewer's concern is valid that proven statements are specific to the idealized setting and do not claim exact transfer to all modern pipelines; the theory generalizes only to methods falling within these assumptions. The core claim is narrow: expected-return maximization weights each outcome's gradient by its sampling probability, a property of the objective, not the architecture. GRPO's softmax LLMs, on-policy sampling, and terminal rewards closely match these assumptions, shrinking the theory-practice gap. Empirical results bridge the rest.
>
> ---
> ## Effect of Noise in Outcome Probability
> IPS-GRPO's ability to estimate outcome probabilities is sensitive to the value of the group size $G$ as the total outcome space grows. Experiments on a 2D grid task varying Horizon length $H$ and Group size $G$ reveal two key behaviors:
>
> - **Sensitivity to Estimation Noise:** As valid outcomes scale, small group sizes lead to poor distribution matching and failed mode discovery. To maintain stability in larger spaces, $G$ must scale proportionally.
> - **Bias from Variance Control:** In massive spaces with small $G$, most outcomes appear once, yielding small empirical probabilities. While the clipping threshold $\epsilon$ prevents reward explosion and controls variance, it introduces bias by weakening the IPS correction. This causes the algorithm to exhibit the early saturation typical of standard GRPO.
> - **Experimental Results**
> The values in the table are: KL Divergence (no of outcomes sampled)
> | $G$ ↓ | $H=8$ [64 outcomes] | $H=16$ [256 outcomes] | $H=32$ [1024 outcomes] | $H=64$ [4096 outcomes] |
> |--------|----------------------|------------------------|-------------------------|-------------------------|
> | 8 | 1.38 (21) | 1.98 (27) | 2.54 (100) | 2.66 (170) |
> | 16 | 0.56 (31) | 1.74 (29) | 2.51 (51) | 2.59 (167) |
> | 32 | 0.25 (43) | 1.73 (30) | 2.31 (76) | 2.63 (138) |
> | 64 | 0.19 (53) | 0.38 (134) | 1.78 (86) | 2.49 (133) |
> | 128 | 0.03 (64) | 0.36 (143) | 0.78 (281) | 2.31 (222) |
> | 256 | 0.07 (64) | 0.09 (254) | 0.44 (326) | 1.81 (269) |
> | 512 | 0.11 (64) | 0.04 (256) | 0.30 (354) | 1.14 (578) |
> | 1024 | 0.10 (64) | 0.09 (256) | 0.15 (786) | 1.03 (652) |
> | 2048 | 0.10 (64) | 0.09 (256) | 0.06 (1024) | 0.32 (1254) |
>
> ---
> ## Applying IPS to Other RL Algorithms
>
> Extending IPS to actor-critic methods like PPO is non-trivial as they rely on learned value or advantage estimates. Because IPS dynamically scales terminal rewards, a "moving-target" mismatch might be introduced between the critic's value estimates and the actor's update objective. Hence, we used GRPO, a critic-free method, as our testbed.
>
> To demonstrate IPS is not tied to GRPO, we applied it to REINFORCE. In this setting, we estimate the outcome probability $\hat{p}(o)$ using empirical frequencies within a rollout batch $\mathcal{B}$ of $N$ trajectories:
>
> $$\hat{p}(o) = \frac{1}{N} \sum_{i=1}^N \mathbb{1}\{ \phi(\tau_i) = o \}$$
>
> Applying the variance-stabilizing threshold $\epsilon$, the IPS-REINFORCE policy gradient estimator is:
>
> $$\nabla_\theta J_{IPS}(\theta) = \frac{1}{N} \sum_{n=1}^N \left[ \sum_{t=0}^{T-1} \nabla_\theta \log \pi_\theta(a_t | s_t) \right] \frac{r(o_n)}{\max(\hat{p}(o_n), \epsilon)}$$
>
> To validate this formulation, we evaluated IPS-REINFORCE against vanilla REINFORCE on the 2D hyper-grid environment extended to a horizon length of 8 (comprising 64 total distinct outcomes), with the following findings:
>
> | Algorithm | KL divergence | Outcomes sampled
> |-----------|----------|---------------|
> | Vanilla REINFORCE | 2.153550 | 1 / 64 |
> | IPS-REINFORCE (Ours) | 0.050460 | 64 / 64 |
>
> Vanilla REINFORCE suffers from severe outcome-level mode collapse, whereas IPS-REINFORCE successfully maintains coverage and accurately recovers the target distribution.
>
> ---
> ## Addressing Quantitative Gaps in High Dimensions
>
> The compression of the quantitative gap is an artifact of using $l_1$ distance in a large-outcome space. In the $n=4$ setting, the terminal state space grows to 4096 possible outcomes, thus, the target probability for any individual state becomes extremely small. Thus, a baseline policy can collapse onto a small fraction of the state space, yet still achieve a low absolute $l_1$ error because the "missing" probability mass per state is tiny. To reconcile this and better quantify the collapse, we evaluated the models using 2 metrics - KL Divergence and Mode Coverage.
>
> | Setting | Algorithm | KL divergence | Outcomes sampled
> |---------|-----------|----------------------|------------------|
> | $n=2$ (64 states) | IPS-GRPO | 0.0062 | 64 / 64 |
> | | FlowRL | 1.062 | 4 / 64 |
> | | GRPO | 2.152 | 2 / 64 |
> | $n=4$ (4096 states) | IPS-GRPO | 0.122 | 4020 / 4096 |
> | | FlowRL | 5.338 | 5 / 4096 |
> | | GRPO | 5.251 | 5 / 4096 |

---

> > ### Author Rebuttal · Reviewer_pS98 · 2026-04-01
> >
> > I am very grateful to the authors for answering my questions. I have raised my rating accordingly.

---

### Official Review · Reviewer_cJVw · 2026-03-11

**Soundness:** 3
**Presentation:** 3
**Significance:** 2
**Originality:** 3
**Overall Recommendation:** 5
**Confidence:** 4

**Summary:**

The paper investigates the loss of diversity and outcome-level mode collapse observed in RL fine-tuned generative models. The authors argue that this mode collapse is structural, arising from the training objective itself rather than from insufficient entropy regularisation or inadequate exploration.

They identify a “rich-get-richer” feedback mechanism in standard policy-gradient-style objectives, where already high-probability outcomes receive disproportionately stronger updates, further concentrating probability mass. To address this, the authors propose a simple method called Inverse Probability Scaling (IPS), which removes the explicit dependence on the selected outcomes’ probabilities in the training objective, thereby breaking this feedback loop.

Building on this idea, the paper introduces IPS-GRPO, a modification of the GRPO algorithm that incorporates inverse probability scaling. Through a series of experiments, the authors demonstrate that IPS-GRPO learns target distributions that better preserve the true distribution’s multimodal structure, rather than collapsing to one or a few highly probable modes.

**Compliance With Llm Reviewing Policy:**

Affirmed.

**Final Justification:**

The authors have addressed the two weakness points I mentioned. From their experiments on REINFORCE, we now have an example of how this method can be applied to different algorithms. Furthermore, the authors have provided rules of thumb for selecting the hyperparameters, $G$ and $\epsilon$. This has cleared my doubts, and I have therefore improved my rating from weak accept to accept. I am not opting for a strong accept because, in my opinion, the significance of the problem addressed in this work is not particularly high.

**Key Questions For Authors:**

1. Can the authors describe how the method could be applied to different algorithms other than GRPO? The paper claims the approach can be easily adapted but this is only illustrated with GRPO.


2. Given the sensitivity of the method to the choice of hyperparameters, G and epsilon. Is there any rule of thumb or principled guideline for selecting appropriate values for these in practice?

**Limitations:**

Yes, technical limitations about the algorithm are discussed. But no societal impacts were discussed.

**Strengths And Weaknesses:**

## Strengths

- The paper is well written, and the problem is clearly demonstrated from an algorithmic perspective, with sufficient explanation and discussion of why the proposed method works.

- The algorithm is evaluated across a wide variety of problem domains, ranging from grid-based tasks, the LLM HypoSpace benchmark (covering multiple task types) and molecular generation.

## Weaknesses

- Although the authors argue that their contribution goes beyond a simple modification of GRPO, it is not entirely clear how the proposed approach would generalise to other RL algorithms beyond the GRPO setting.

- The proposed method also appears quite sensitive to the hyperparameters—particularly the number of groups, G and the epsilon term. The sensitivity to epsilon is especially concerning, since this quantity is intended to serve merely as a numerical safeguard against division by zero for outcomes with infinitesimal probabilities.

---

> ### Author Rebuttal · Authors · 2026-03-31
>
> ## Applying IPS to other RL algorithms
> Extending IPS (designed to be a drop-in modification for critic-free methods) to actor-critic methods (like PPO) is nontrivial, as they  rely on learned value function for advantage estimates. Because IPS dynamically scales the terminal reward based on current policy outcome frequencies, a severe moving-target mismatch could be introduced between the critic's value target and the actor's update objective. Thus GRPO was chosen as the primary testbed, as its grouped on-policy sampling allows for a clean empirical estimation of outcome probabilities. To demonstrate that IPS is not mechanically tied to GRPO, we apply it to vanilla REINFORCE. In standard REINFORCE, we sample a batch of $N$ trajectories $\mathcal{B} = \{\tau_1, \dots, \tau_N\}$, each yielding an outcome $o_n = \phi(\tau_n)$ and reward $r(o_n)$.
>
> Applying the IPS requires replacing the standard reward with the inverse-scaled reward $\tilde{r}(o_n) = \frac{r(o_n)}{p_\theta(o_n)}$. While GRPO estimates this probability using group frequencies, in REINFORCE, we estimate $\hat{p}(o)$ by computing its empirical frequency across the current rollout batch $\mathcal{B}$:
>
> $$\hat{p}(o) = \frac{1}{N} \sum_{i=1}^N \mathbb{1}\{ \phi(\tau_i) = o \}$$
>
> Applying the variance-stabilizing clipping threshold $\epsilon$, the IPS-REINFORCE policy gradient estimator seamlessly becomes:
>
> $$\nabla_\theta J_{IPS}(\theta) = \frac{1}{N} \sum_{n=1}^N \left[ \sum_{t=0}^{T-1} \nabla_\theta \log \pi_\theta(a_t | s_t) \right] \frac{r(o_n)}{\max(\hat{p}(o_n), \epsilon)}$$
>
> To validate this formulation, we evaluated IPS-REINFORCE against vanilla REINFORCE on the 2D hyper-grid environment extended to a horizon length of 8 (comprising 64 total distinct outcomes), with the following findings on KL divergence between the learned terminal-state distribution and the target reward distribution and Unique outcomes sampled by the trained policy.
>
> | Algorithm | KL divergence | Outcomes sampled|
> |---|---|---|
> | Vanilla REINFORCE | 2.15 | 1 / 64 |
> | IPS-REINFORCE (Ours) | 0.05 | 64 / 64 |
>
> we can see that vanilla REINFORCE suffers from severe outcome-level mode collapse, whereas IPS-REINFORCE successfully maintains coverage and accurately recovers the target distribution.
>
> ---
> ## Choice of Hyperparameter
>
> IPS-GRPO's performance is governed by the accuracy of the empirical outcome frequency estimation, $\hat{p}(o)$. The perceived "sensitivity" arises when the parameters fail to provide sufficient statistical resolution.
>
> 1. Group Size ($G$) Heuristic - $G$ must be large enough to sample duplicate outcomes within a batch; otherwise, $\hat{p}(o)$ degrades to a uniform distribution of singletons. If the true number of modes is huge, $G$ acts as a "resolution budget" defining the maximum number of modes the policy can actively balance per update. Thus, it is a good rule of thumb to set $G$ on the order of the expected number of optimal outcomes.
> Experimentally, we see that, in an environment with 256 outcomes ($H=16$), $G=16$ yields poor coverage and high KL divergence (1.74). However, as $G \to 256$, KL divergence drops to 0.08, and the model recovers 254 of the 256 modes.
> The values in the table are: KL Divergence (no of outcomes sampled)
> | G↓ | H=8 [64 outcomes] | H=16 [256 outcomes] | H=32 [1024 outcomes] | H=64 [4096 outcomes] |
> |---|---|---|---|---|
> | 8 | 1.38 (21) | 1.98 (27) | 2.54 (100) | 2.66 (170) |
> | 16 | 0.56 (31) | 1.74 (29) | 2.51 (51) | 2.59 (167) |
> | 32 | 0.25 (43) | 1.73 (30) | 2.31 (76) | 2.63 (138) |
> | 64 | 0.19 (53) | 0.38 (134) | 1.78 (86) | 2.49 (133) |
> | 128 | 0.03 (64) | 0.36 (143) | 0.78 (281) | 2.31 (222) |
> | 256 | 0.07 (64) | 0.09 (254) | 0.44 (326) | 1.81 (269) |
> | 512 | 0.11 (64) | 0.04 (256) | 0.30 (354) | 1.14 (578) |
> | 1024 | 0.10 (64) | 0.09 (256) | 0.15 (786) | 1.03 (652) |
> | 2048 | 0.10 (64) | 0.09 (256) | 0.06 (1024) | 0.32 (1254) |
>
>
> 2. Probability Clipping ($\epsilon$) Heuristic: $\epsilon$ controls the variance of updates for rare outcomes. An outcome sampled exactly once has an empirical probability of $1/G$. Setting $\epsilon \ll 1/G$ artificially inflates the reward of singletons, making clipping superfluous causing the instability noted in our ablations. Conversely, $\epsilon \gg 1/G$ aggressively suppresses the IPS correction, inducing mode collapse. Thus, a good rule of thumb would be to start $\epsilon \approx 1/G$ and if training exhibits high variance, incrementally increase $\epsilon$ (e.g., to $2/G$ or $3/G$) to clip the rarest outcomes until stability is achieved.

---

> > ### Author Rebuttal · Reviewer_cJVw · 2026-04-06
> >
> > The authors have addressed the two weakness points I mentioned. From their experiments on REINFORCE, we have an example of how this method could be applied to a different algorithm. Furthermore, the authors have provided rules of thumb for selecting the hyperparameters, $G$ and $\epsilon$.
> >
> > I will adjust my score to 5.

---

### Official Review · Reviewer_Lec2 · 2026-03-13

**Soundness:** 2
**Presentation:** 3
**Significance:** 2
**Originality:** 2
**Overall Recommendation:** 5
**Confidence:** 3

**Summary:**

The work highlights a mode collapse issue with the standard RL reward maximization objective and proposes weighting the standard RL objective by the inverse probability of different outcomes to ensure that outcomes are sampled in proportion to their reward.

**Compliance With Llm Reviewing Policy:**

Affirmed.

**Final Justification:**

Authors addressed my concerns in the rebuttal. I have increased the score accordingly.

**Key Questions For Authors:**

## Questions
- Do the authors have any thoughts on additive vs. multiplicative reweightings of rewards to combat the issue of diversity collapse. Methods like [1] use additive factors, and I believe they have the advantage of being more stable and not requiring any clipping. However, they do require deciding a threshold $\tau$  for the reward beyond which all generations should have identical probabilities.
- Is there any specific reason why the FlowRL baseline performs similarly to GRPO in terms of the number of modes found? Since FlowRL was designed to mitigate diversity collapse via distribution matching, I would have expected it to perform better.
- The theoretical analysis predicts mode collapse under expected-return maximization, yet models like DeepSeek-R1 were trained with GRPO for thousands of RL iterations without any loss of diversity. Is this because the long chain-of-thought horizon makes the outcome space so vast that collapse is slower (exponentially?) in practice? Or is it because the pretrained base model already had sufficient coverage across solution strategies, providing a strong enough regularization against collapse? More broadly, do the authors have any intuition for how horizon length interacts with the rate of diversity loss predicted by Theorem 3.1?

**Limitations:**

yes

**Strengths And Weaknesses:**

## Strengths
- The paper is well written and easy to follow.
- The theoretical analysis done offers a complementary perspective to related work [1] by characterizing collapse through gradient flow dynamics rather than properties of the optimal solution.
- The proposed fix is simple and minimal and can be easily incorporated in RL training pipelines

## Weaknesses and Suggestions

- It would be interesting to see the effect of the proposed approach on more practical tasks such as math reasoning or code, where the horizon lengths are much longer. Unlike the tasks considered in the paper, the notion of diversity is a bit loose in such tasks.
- Methods like [1] and [2] seem like pretty relevant baselines. [2] also incorporates an inverse-propensity-like reweighting factor in the objective. It might help to compare against these methods.
- In cases where the output spaces are large, it is unclear how the inverse propensity weighting would be helpful. From what I understand, if the policy outputs a different answer for each of the G generations, then all the generations are scaled equally, thus resulting in the same gradient as the original case without any reweighting.

[1]: GX-Chen, Anthony, et al. "KL-Regularized Reinforcement Learning is Designed to Mode Collapse." ICLR 2026.

[2]: Chow, Yinlam, et al. "Inference-aware fine-tuning for best-of-n sampling in large language models." ICLR 2025.

---

> ### Author Rebuttal · Authors · 2026-03-31
>
> ## Math Reasoning & Code Generation
> In math/code tasks, semantically equivalent outcomes are syntactically distinct. Applying IPS here requires replacing empirical frequency with density estimators or learned similarity kernels over a latent space which is a non-trivial extension adding overhead to the current method. However, this is not a weakness of the core IPS principle which remains fully valid once outcome equivalence is appropriately defined.
>
> ---
> ## Behavior in Large Outcome Spaces
> When all $G$ sampled outcomes are unique, IPS reduces to uniform scaling. This is because the empirical estimator lacks the resolution to detect outcome-frequency imbalance. Additional hyper-grid experiments varying group size $G$ and outcome size $H$, confirm IPS-GRPO converges reliably when $G$ is comparable to the effective reachable outcomes.
> | $G$ ↓ | $H=8$ [64 outcomes] | $H=16$ [256 outcomes] | $H=32$ [1024 outcomes] | $H=64$ [4096 outcomes] |
> |-------:|--------------------:|----------------------:|-----------------------:|-----------------------:|
> | 32     | 0.25 (43)           | 1.73 (30)             | 2.31 (76)              | 2.63 (138)             |
> | 128    | 0.03 (64)           | 0.36 (143)            | 0.78 (281)             | 2.31 (222)             |
> | 512    | 0.11 (64)           | 0.04 (256)            | 0.30 (354)             | 1.14 (578)             |
> | 2048   | 0.10 (64)           | 0.09 (256)            | 0.06 (1024)            | 0.32 (1254)            |
>
> ---
> ## [1],[2] additive vs. multiplicative
> [1] attributes collapse to KL regularization and modifies the penalty. 3.1 shows collapse arises from the expected-return objective itself, independent of the regularizer; IPS targets this directly. [2] uses inverse-propensity for optimizing best-of-N (a max-statistic), concentrating probability on the top outcome. IPS instead yields reward-proportional distributions, preserving multimodal coverage.
>
> **On additive vs. multiplicative:** $1/p(o)$ scaling algebraically cancels the $p(o)$ factor in the gradient , structurally removing the rich-get-richer loop rather than merely opposing it. It also yields a specific stationary solution $\pi_i ∝ r_i$, reflecting the reward landscape without manual thresholds. Additive methods introduce a restoring force but do not eliminate the destabilizing mechanism, and require choosing a reward cutoff above which outcomes are treated uniformly. We acknowledge additive methods offer practical stability advantages without clipping, robustness to noisy density estimates, and may be preferable with sparse/binary rewards. We view these as complementary paradigms and will add a discussion in the revision.
>
> ---
> ## FlowRL Baseline Performance
>
> FlowRL struggles with outcome-level diversity in our settings due to three fundamental mathematical issues:
>
> **1. Trajectory vs. Outcome (Tree vs. DAG):** FlowRL optimizes diversity over trajectories ($y$). In our DAG-like tasks, multiple paths collapse to one outcome ($o$). FlowRL can satisfy its objective by exploring diverse paths to a single mode, failing to prevent outcome-level collapse.
>
> **2. Equal-Reward Degeneracy:** With equal binary rewards ($r_0$), the target
>
> $$\tilde{\pi}(y|x) = \frac{e^{\beta r_0}\pi_{\mathrm{ref}}(y|x)}{Z_\phi(x)}$$
>
> simplifies since the partition function becomes $Z_\phi(x) = e^{\beta r_0}$. The target collapses to $\tilde{\pi}(y|x) = \pi_{\mathrm{ref}}(y|x)$. FlowRL defaults to minimizing $D_{\mathrm{KL}}(\pi_\theta \parallel \pi_{\mathrm{ref}})$ instead of actively covering modes.
>
> **3. Length Bias:** FlowRL's length-normalized loss minimizes the squared error of average token log-probability against a state-dependent constant. Because longer trajectories accumulate more negative terms, FlowRL systematically biases toward shorter trajectories where average per-token log-probability is easier to maximize. We will add an Appendix section detailing these mechanisms.
>
> ---
>
> ## DeepSeek-R1 and Horizon Length
>
> DeepSeek-R1's diversity under GRPO does not contradict Theorem 3.1. The collapse is mathematically inevitable but practically frozen on the training timescale by two factors:
>
> **Strong Prior:** The KL penalty against a highly capable, well-trained base model provides continuous resistance to the collapse pressure.
>
> **Horizon Scaling:** We can approximate the log-ratio drift from Theorem 3.1 to formalize this. Assuming a horizon $H$, action space $|A|$, and $N(i)$ valid paths to outcome $i$, the per-path probability is $|A|^{-H}$. The drift is:
>
> $$\frac{d}{dt}\log\frac{p_i(t)}{p_j(t)} \approx a \cdot |A|^{-H} \cdot (N(i) - N(j))$$
>
> For Chain-of-Thought (CoT), $H$ is astronomically large, making $|A|^{-H}$ infinitesimal. The gradient updates driving the positive feedback loop are microscopic. As RL is applied to shorter reasoning tasks or with weaker KL constraints, this underlying collapse will surface. We will add this formal derivation to the Appendix to contextualize long-horizon CoT models.

---

> > ### Author Rebuttal · Reviewer_Lec2 · 2026-04-03
> >
> > I thank the authors for addressing my questions. I believe that comparisons with relevant baselines such as [1] and [2] would help clarify the benefits of IPS relative to similar methods. I would like to retain my score.

---

> > > ### Author Response · Authors · 2026-04-06
> > >
> > > We thank the reviewer for the follow-up.
> > >
> > > As requested, we have run experiments comparing IPS-GRPO against [1] (MARA) and [2] (BoN-RLB, Algorithm 3) on the Hypospace benchmark with an LLM backbone. Results below report the percentage of distinct correct outcomes discovered:
> > >
> > > | Method       | Causal Inference    | 3D Reconstruction   | Boolean/DNA         |
> > > |--------------|---------------------|---------------------|---------------------|
> > > | GRPO         | 16.02% ± 0.35%      | 31.61% ± 0.01%      | 9.50% ± 0.00%       |
> > > | FlowRL       | 12.82% ± 0.05%      | 26.85% ± 0.37%      | 10.74% ± 0.00%      |
> > > | MARA         | 21.47% ± 0.39%      | 63.05% ± 0.52%      | 28.10% ± 0.39%      |
> > > | BoN-RLB      | 16.03% ± 0.25%      | 28.95% ± 0.35%      | 9.92% ± 0.00%       |
> > > | **IPS-GRPO** | **43.91% ± 1.39%**  | **90.00% ± 1.24%**  | **60.74% ± 0.78%**  |
> > >
> > > IPS-GRPO consistently and substantially outperforms both baselines across all three tasks of Hypospace. We will incorporate these comparisons into the revised manuscript.

---

### Decision · Program_Chairs · 2026-04-30

**Decision:**

Accept (regular)

**Comment:**

This paper studies outcome-level mode collapse in reinforcement learning, particularly in settings where multiple high-quality terminal outcomes exist and diversity is essential. Overall, a fundamental problem discussed by this paper is that standard expected-return maximization inherently biases policies toward a small subset of outcomes, even when many solutions are equally valid. The authors argue that this phenomenon is not merely due to insufficient exploration or regularization, but is instead a structural property of the objective itself.

Reviewers agreed that the paper provides a clear and compelling perspective on this issue. The theoretical analysis is a strong component of the work, offering a clean characterization of learning dynamics that explains why probability ratios between outcomes diverge over time. This insight is both intuitive and non-trivial, and helps clarify longstanding empirical observations in RL and LLM post-training. Overall, this submission's critical contribution consists of identifying the cause of outcome-level mode collapse at the level of the objective, and proposing a minimal correction via inverse probability scaling (IPS) that directly targets this mechanism.

Empirically, the proposed IPS-GRPO method is evaluated across a diverse set of domains, including controlled grid environments, reasoning benchmarks, and molecular generation. The results consistently show improved coverage of high-reward outcomes and reduced collapse, while maintaining or improving performance. In particular, experiments on the HypoSpace benchmark demonstrate substantial gains in recovery rate and diversity, and the molecular optimization results further highlight the practical impact of the approach.

At the same time, reviewers noted several limitations. The theoretical analysis is based on idealized assumptions, and its applicability to more complex RL pipelines remains partially indirect. The method relies on empirical estimation of outcome probabilities, which becomes challenging in large or continuous outcome spaces. In addition, most experiments focus on GRPO-style policy gradient methods, and extending the approach to actor-critic frameworks is non-trivial. Some concerns were also raised regarding sensitivity to hyperparameters such as group size and clipping.

These issues were discussed during the rebuttal phase. The authors provided additional experiments, including comparisons to relevant baselines and extensions beyond GRPO (e.g., REINFORCE), as well as clarifications on scalability and hyperparameter selection. These responses addressed most of the reviewers’ concerns and strengthened confidence in the robustness of the approach.

Overall, the paper presents a clear and impactful idea, supported by solid theoretical reasoning and convincing empirical results. Despite some limitations in generality and scalability, the work offers a valuable new perspective on objective design in reinforcement learning and provides a simple, practical method that is likely to influence future research. My recommendation is to accept.